# FlowCloud: Learning Continuous Spatiotemporal Dynamics from Unpaired Sparse Point Cloud Snapshots

**Yinbo Liu** [* 1 2]  **Keyang Ye** [* 3]  **Wenshan Sun** [4]  **Handi Gao** [5]  **Tian Tian** [1 2 †]

## Abstract

Reconstructing unified continuous dynamics from sparse, non-contiguous, and unpaired point cloud snapshots remains a fundamental challenge in spatiotemporal analysis for computer vision and developmental biology. Existing methods, including scene flow and Optimal Transport-based approaches, are limited either by their explicit reliance on point-wise correspondences or by cumulative errors arising from frame-to-frame propagation and temporal discontinuity, and they often have limited ability to model multi-attribute dynamics such as gene expression and population changes. We propose FlowCloud, a variational Neural Ordinary Differential Equation (Neural ODE) generative framework. FlowCloud aggregates information from all observed time points into a joint latent representation that initializes a Neural ODE $z(t)$ enabling continuous spatiotemporal evolution modeling while mitigating propagation-induced errors and preserving temporal consistency. Training is performed without predefined correspondences using a multi-faceted objective with complementary roles: Sinkhorn distance for global distribution alignment, Chamfer distance for local geometric consistency, trajectory regularization to encourage smooth and physically plausible dynamics, and supervised losses for multi-attribute prediction. Experiments on human motion and developmental biology datasets demonstrate improved interpolation accuracy and promising short-term extrapolation performance. By unifying geometry, attributes, and

[1] School of Computer Science, Wuhan University, Wuhan, China [2] School of Artificial Intelligence, Wuhan University, Wuhan, China [3] College of Life Sciences, Wuhan University, Wuhan, China [4] School of Public Health, Wuhan University, Wuhan, China [5] School of Mathematics and Statistics, Wuhan University, Wuhan, China. Correspondence to: Yinbo Liu <yinboliu@whu.edu.cn>, Tian Tian <tiantian87@whu.edu.cn>.

*Proceedings of the 43rd International Conference on Machine Learning*, Seoul, South Korea. PMLR 306, 2026. Copyright 2026 by the author(s).

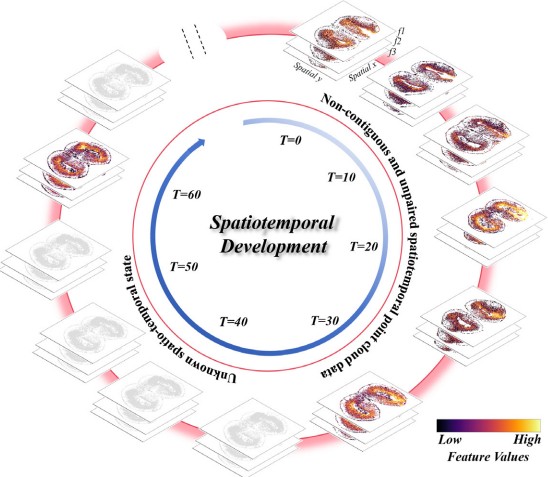

*Figure 1.* Motivation: Inferring Spatiotemporal Dynamics Trajectories from Sparse Transcriptomic Snapshots with Varying Cell Counts.

population dynamics within a continuous latent framework, FlowCloud offers a novel and robust solution for continuous dynamic reconstruction from unstructured spatiotemporal observations.

## 1. Introduction

Reconstructing unified continuous dynamics from sparse, non-consecutive, and unpaired point cloud snapshots is a core challenge in developmental biology (Qiu et al., 2024; Wang et al., 2025) and computer vision (Liu et al., 2019b). In biological systems, this capability is essential for inferring cellular differentiation trajectories from discrete spatial omics measurements to understand fundamental processes like morphogenesis and pathogenesis (Figure 1) (Klein et al., 2025; Qiu et al., 2024; Peng et al., 2025; Wei et al., 2022). In vision, it underlies the recovery of complete motion sequences from fragmented visual observations (Bogo et al., 2017; Lin et al., 2025b; Liu et al., 2019a). Beyond simple alignment, the goal is to learn a generative model that captures the underlying data distribution for retrospective trajectory reconstruction. Specifically, FlowCloud acts as

a non-causal smoother designed to understand a biological process that has already occurred from discrete historical slices, rather than performing real-time future forecasting. Given sparse, discontinuous, and unpaired data, such a model must reconstruct smooth spatiotemporal trajectories from a few keyframes, while supporting accurate interpolation and extrapolation (Chen et al., 2018).

Despite recent progress, existing approaches exhibit fundamental limitations. First, scene flow methods (Puy et al., 2020; Cheng & Ko, 2023; Liu et al., 2024; Lin et al., 2025a), such as FlowNet3D (Liu et al., 2019a) and PointPWC-Net (Wu et al., 2020; Sun et al., 2018), rely on dense, strictly paired temporal sequences (Cheng & Ko, 2023; Liu et al., 2019b), limiting their applicability to unpaired and sparse settings. Second, Optimal Transport (OT) frameworks (Peyré & Cuturi, 2019; Schiebinger et al., 2019; Lin & Caesar, 2024) compute discrete matchings rather than continuous velocity fields $\frac{dz}{dt} = f(z, t)$. Consequently, Optimal Transport–based methods, including recent approaches such as Moscot (Klein et al., 2025), lack the capacity for smooth interpolation and struggle with complex population behaviors like proliferation and death (Peng et al., 2025). Third, propagation-based Neural CDE/ODE models (Peng et al., 2025) typically assume Markovian transitions $(z(t_i) \rightarrow z(t_{i+1}))$, making them vulnerable to cumulative errors and drift over long horizons. Finally, most existing methods primarily focus on geometric alignment or trajectory inference, thereby limiting their ability to jointly model multiple attributes, including gene expression, cell types, and population dynamics (including point birth, death, and merging) (Lin & Caesar, 2024; Wang et al., 2022).

To overcome these limitations, we propose FlowCloud, a variational Neural ODE generative framework for unified spatiotemporal modeling, which enables distribution-level modeling under sparse and unpaired observations. Unlike propagation-based methods that rely on frame-to-frame transitions, FlowCloud aggregates deep spatial features from all snapshots into a single global spatiotemporal context vector. This initializes the state $z(0)$ of a Neural ODE, enabling the learning of a unified velocity field $f(z, t)$ that governs the entire evolution (Figure 2). By treating all time points with equivalent importance, we avoid cumulative errors and the need for point-wise correspondences. To ensure physical plausibility, we introduce a composite loss function that enforces both global distribution alignment and local geometric consistency, while incorporating trajectory consistency and acceleration regularization. By evolving $z(t) = \text{ODESolve}(z(0), f, t)$, we decouple dynamics from propagation errors. A decoder then maps $z(t)$ back to physical space, simultaneously generating (1) geometric morphology, (2) discrete cell types, (3) continuous gene expression profiles, and (4) population dynamics, including point growth and disappearance.

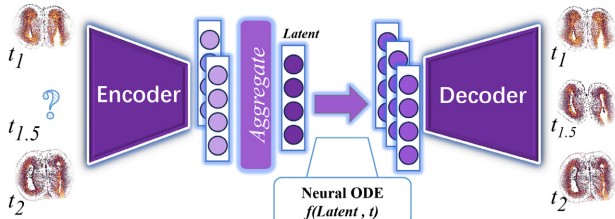

*Figure 2.* Task and Architecture Diagram of the Model.

Our main contributions are summarized below. We propose FlowCloud, a variational Neural ODE generative framework that learns a continuous spatiotemporal dynamics model from a small number of sparse, non-consecutive, and unpaired point cloud snapshots. FlowCloud addresses both the spatiotemporal discontinuity of discrete methods such as Optimal Transport and the cumulative error issue of propagation-based continuous models such as recurrent ODEs. It enables comprehensive generative modeling by unifying the co-evolution of geometric structure, discrete labels, continuous features, and population dynamics within a single end-to-end framework. To support training without explicit correspondences, we design a multi-faceted loss function with complementary roles: the Sinkhorn distance (Optimal Transport) (Peyré & Cuturi, 2019) for global distribution alignment and the Chamfer distance for local geometric fidelity; trajectory consistency and acceleration regularization encourage physically plausible temporal dynamics; and additional supervised losses enable the prediction of multiple attributes, including gene expression and population dynamics. FlowCloud significantly outperforms existing state-of-the-art baselines (Klein et al., 2025; Cheng & Ko, 2023; Lin et al., 2025b) on diverse and challenging datasets, including simulated data, human motion (D-FAUST) (Bogo et al., 2017), and spatiotemporal transcriptomics data (Wei et al., 2022; Wang et al., 2025).

## 2. Related Works

Understanding and modeling dynamic processes from 3D point cloud sequences is a fundamental problem in computer vision and developmental biology. The problem addressed in this work lies at the intersection of three research directions: dense scene flow estimation, discrete-time point cloud registration/optimal transport, and continuous-time dynamics modeling.

Scene flow estimation aims to compute a dense 3D displacement vector field, typically between two consecutive and paired point clouds (Liu et al., 2019a). Pioneering works like FlowNet3D (Liu et al., 2019a) and PointPWC-Net (Wu et al., 2020; Sun et al., 2018) adopt a coarse-to-fine hierarchical structure to regress the flow. Subsequent work has largely focused on improving refinement, robustness, and

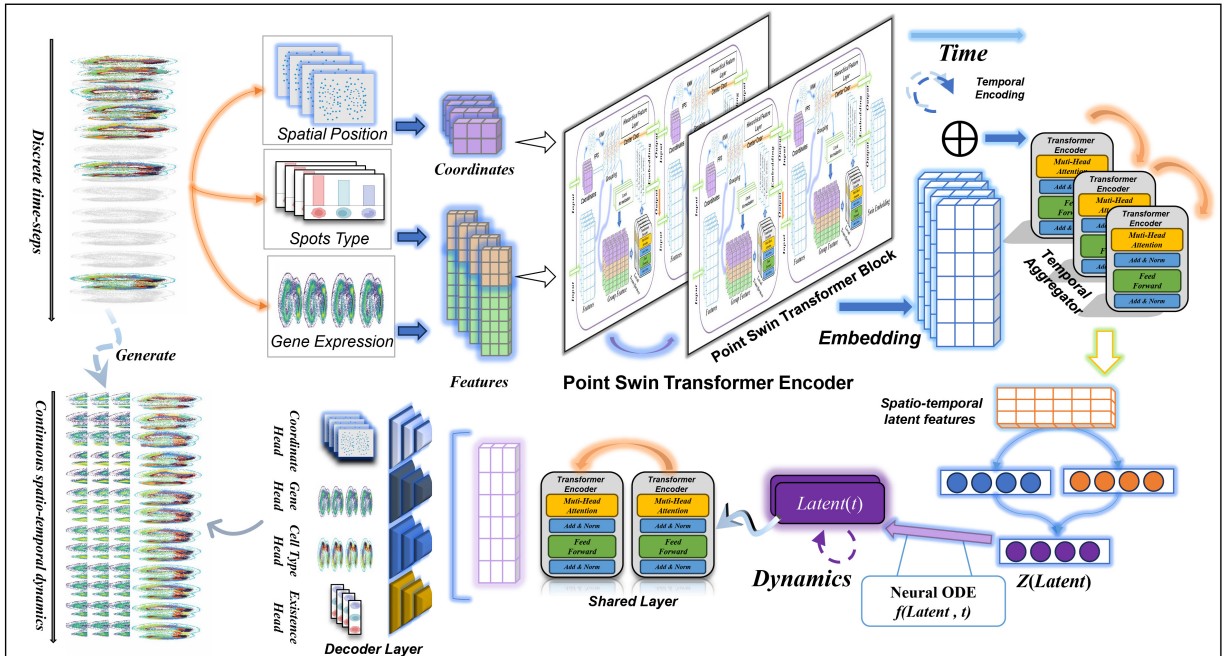

*Figure 3.* Overview of the FlowCloud framework for learning continuous spatiotemporal dynamics from sparse point cloud snapshots.

efficiency (Teed & Deng, 2021; Wei et al., 2021; Wang et al., 2021; Cheng & Ko, 2023; Liu et al., 2024; Lin et al., 2025a). Meteornet (Liu et al., 2019b) is an early attempt to process sequences longer than two frames, but still requires a dense input stream. A common limitation of these methods is their reliance on dense, continuous, and strictly paired input frames (Cheng & Ko, 2023; Liu et al., 2019b; He et al., 2022). This makes them unsuitable for sparse, non-consecutive, and unpaired snapshots. More recently, Neural Trajectory Prior (NTP) (Wang et al., 2022) infers continuous trajectories from unpaired observations via learned neural priors with implicit smoothness and low-rank regularization, but is not designed for joint modeling multiple attributes.

A second line of work relaxes the requirement of strictly consecutive observations but focuses on computing discrete and static mappings. Classical algorithms such as Iterative Closest Point (ICP) (Lin & Caesar, 2024) iteratively estimate optimal rigid transformations between point clouds. Recent deep learning-based methods, including VoteFlow (Lin et al., 2025b) and ICP-Flow (Lin & Caesar, 2024), incorporate local rigidity priors to improve non-rigid registration. However, these methods are designed for registration rather than for modeling generative non-rigid dynamics, and therefore do not produce continuous trajectories over time. Optimal Transport (OT) (Peyré & Cuturi, 2019) has also emerged as a powerful tool for aligning distributions, particularly in spatial transcriptomics (Schiebinger et al., 2019). Methods such as Spateo (Qiu et al., 2024) and Moscot (Klein et al., 2025) utilize (Fused) Gromov-Wasserstein OT to align

spatial transcriptomics data across time points or modalities. While state-of-the-art for mapping discrete snapshots, OT-based approaches remain static: they compute optimal couplings between observations but do not learn an underlying continuous-time dynamic function governing temporal evolution.

Beyond discrete mapping, recent works explore continuous-time modeling using neural differential equations. In biology, TrajectoryNet (Tong et al., 2020) and stVCR (Peng et al., 2025) employ ODEs to propagate cellular dynamics between consecutive single-cell and spatial observations. In computer vision, EulerFlow (Vedder et al., 2025) models scene dynamics as a continuous spatiotemporal velocity field from an Eulerian perspective, while SPCMNET (He et al., 2022) introduces a recurrent architecture for sequential scene flow estimation. Despite leveraging continuous-time representations, these approaches remain limited in the context of our problem setting. EulerFlow and SPCMNET are primarily designed for geometry-only motion reconstruction and do not support joint generative modeling of multi-attribute dynamics or variable point cardinality. In contrast, propagation-based models (e.g., TrajectoryNet and stVCR) rely on frame-by-frame transitions, making them susceptible to cumulative errors and long-term drift.

## 3. Method

We introduce FlowCloud, a generative framework designed to learn unified continuous spatiotemporal dynamics from

sparse, non-consecutive, and unpaired point cloud snapshots (Figure 2). The full framework is visualized in Figure 3. FlowCloud is formulated as a variational neural ordinary differential equation (VAE–ODE) model (Kingma & Welling, 2014; Chen et al., 2019), which represents the entire temporal evolution through a latent trajectory $z(t)$. The model integrates a geometric encoder for snapshot-level representation learning, a global context–conditioned dynamics kernel that parameterizes the latent ODE, and a multi-head attention–based decoder for reconstructing observations across time.

### 3.1. Problem Formulation and Architectural Overview

Our input is a discrete set of sparsely observed point cloud snapshots $\mathcal{S} = \{S_{t_i}\}_{i=1}^N$ at $t_1 < t_2 < \cdots < t_N$. Each snapshot $S_{t_i} \in \mathbb{R}^{N_i \times D_{\text{in}}}$ contains $N_i$ points, each with $D_{\text{in}}$-dimensional features. Our objective is to learn a continuous latent dynamics function $z(t) : \mathbb{R} \to \mathbb{R}^{D_z}$ that represents the global evolutionary state of the system (Chen et al., 2018; 2019) . Given $z(t)$, a generative decoder $\mathcal{D}$ can reconstruct the complete point cloud state $\hat{S}(t) = \mathcal{D}(z(t))$ at any arbitrary continuous time $t$. FlowCloud's architecture comprises four core components (Figure 3): (1) A Point Swin Transformer snapshot encoder ($\mathcal{E}_{\text{point}}$) to independently encode each $S_{t_i}$ into a latent vector $z_i \in \mathbb{R}^{D_z}$. (2) A sequence Transformer ($\mathcal{E}_{\text{seq}}$) to aggregate all $\{z_i\}$ into a single global context vector $c$. (3) A variational dynamics kernel ($\mathcal{F}_{\text{ODE}}$) that uses $c$ to parameterize the initial state $z(t_0)$ of a Neural ODE, $\frac{dz}{dt} = f_\theta(z(t), t)$. (4) A multi-head decoder ($\mathcal{D}$) to map $z(t)$ back to the full point cloud, generating its geometry, attributes, and population.

### 3.2. Spatiotemporal Encoder and VAE Dynamics

The encoder's goal is to compress the entire observed sequence into the single initial state $z(t_0)$. This process is sequential: we first encode each snapshot spatially, then aggregate these encodings temporally.

#### 3.2.1. SNAPSHOT ENCODING: SWIN HIERARCHICAL FEATURE EXTRACTION

We use a **Swin Hierarchical Point Cloud Encoder** as $\mathcal{E}_{\text{point}}$. This module merges the hierarchical sampling paradigm of PointNet++ (Qi et al., 2017) with the local attention mechanism from the Swin Transformer (Figure 4). Let the input snapshot be $S_{t_i} = (P^{(0)}, F^{(0)})$, where $P^{(0)} \in \mathbb{R}^{N_i \times D_{\text{spatial}}}$ representing spatial positions and $F^{(0)} \in \mathbb{R}^{N_i \times D_{\text{feature}}}$ representing features (e.g., gene expressions and cell types). The encoder $\mathcal{E}_{\text{point}}$ is a stack of $L$ hierarchical layers. Each layer $l$ transforms $(P^{(l)}, F^{(l)})$ to $(P^{(l+1)}, F^{(l+1)})$ via three steps: 1. **Sampling**: A subset of $N_{l+1}$ centroids $P^{(l+1)}$ is sampled from $P^{(l)}$ via Farthest Point Sampling (FPS):

$$P^{(l+1)} = \text{FPS}(P^{(l)}, N_{l+1}) \tag{1}$$

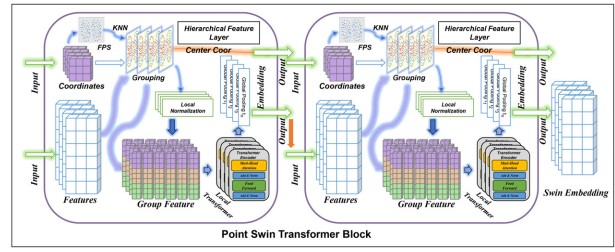

*Figure 4.* Point Swin Transformer Block for Hierarchical Local Feature Extraction.

2. **Grouping**: For each centroid $p' \in P^{(l+1)}$, we find its $k$-Nearest Neighbors ($k$NN) neighborhood $\mathcal{N}(p')$ from the original set $P^{(l)}$. 3. **Local Attention**: The features of the neighborhood, $\mathbf{F}_{\mathcal{N}(p')}$, are concatenated with their relative coordinates $\Delta P_{\mathcal{N}(p')}$ and processed by a local Transformer block to compute the new feature $f'^{(l+1)}_{p'}$ corresponding to the centroid $p'$:

$$f'^{(l+1)}_{p'} = \text{MaxPool}\Big(\text{LocalTransformer}\big( \\ \text{Concat}(\Delta P_{\mathcal{N}(p')}, \mathbf{F}_{\mathcal{N}(p')})\big)\Big) \tag{2}$$

This process is repeated hierarchically. The final latent vector $z_i$ for the snapshot $S_{t_i}$ is obtained by applying global max pooling to the features of the last layer, $F^{(L)}$:

$$z_i = \text{GlobalMaxPool}(F^{(L)}) = \max_k \left\{ f_k^{(L)} \right\} \tag{3}$$

#### 3.2.2. SEQUENCE CONTEXT AND VARIATIONAL INITIAL STATE

The sequence of latent vectors $\mathbf{Z} = [z_1, \ldots, z_N] \in \mathbb{R}^{N \times D_z}$ is augmented with standard positional encodings and fed into a sequence Transformer encoder $\mathcal{E}_{\text{seq}}$ (Vaswani et al., 2017) to apply global self-attention across all time points:

$$\mathbf{H} = \mathcal{E}_{\text{seq}}(\mathbf{Z} + \mathbf{P}_{\text{time}}) \tag{4}$$

The output $\mathbf{H} \in \mathbb{R}^{N \times D_z}$ represents the temporally-contextualized features of all snapshots. We aggregate it into a single global context vector $c$ via mean-pooling, $c = \frac{1}{N} \sum_{i=1}^N \mathbf{H}_i$. From a biological perspective, this global context acts as a developmental blueprint. Tissue morphogenesis is a globally coordinated process guided by systemic signals; thereby, extracting this macroscopic signature enables the model to oversee the overarching spatiotemporal dynamics prior to inferring individual time points, mitigating the reliance on error-prone Markovian transitions. This context $c$ defines the posterior distribution of the initial latent state $q(z(t_0)|\mathcal{S}) = \mathcal{N}(\mu_{z_0}, \sigma_{z_0}^2)$, with parameters mapped from $c$:

$$[\mu_{z_0}, \log \sigma_{z_0}^2] = \text{MLP}(c) \tag{5}$$

We sample the initial state $z(t_0)$ for the ODE integration using the reparameterization trick (Kingma & Welling, 2014):

$$z(t_0) = \mu_{z_0} + \epsilon \cdot \exp\left(0.5 \cdot \log \sigma_{z_0}^2\right), \quad \epsilon \sim \mathcal{N}(\mathbf{0}, \mathbf{I}) \quad (6)$$

### 3.2.3. CONTINUOUS LATENT DYNAMICS WITH NEURAL ODE

The continuous evolution of the system is governed by a Neural ODE (Chen et al., 2018). Because cellular differentiation and migration are strictly continuous processes in physical time, modeling the latent space with a Neural ODE physically aligns with this continuous flow, accurately representing the underlying continuous vector field of cell fate transitions. Our latent velocity network $f_\theta$ is explicitly conditioned on both the latent state $z(t)$ and the time $t$, $\frac{dz(t)}{dt} = f_\theta(z(t), t)$, which we implement as $f_\theta(z(t), t) = \text{MLP}_{\text{vel}}(\text{Concat}(z(t), \phi_t(t)))$, where $\phi_t$ is a temporal MLP encoder. The latent state at any time is defined by integrating this ODE:

$$z(t) = z(t_0) + \int_{t_0}^{t} f_\theta(z(\tau), \tau)\, d\tau \quad (7)$$

### 3.3. Generative Point Cloud Decoder

The decoder $\mathcal{D}$ is an attention-based architecture that maps the continuous latent state $z(t)$ back to a full physical state consisting of $N_{\text{max}}$ points. To this end, it employs a set of $N_{\text{max}}$ learnable **Queries** $\mathbf{Q} \in \mathbb{R}^{N_{\text{max}} \times D_z}$, which act as latent slot representations for the output points (Locatello et al., 2020). The latent state $z(t)$ is linearly projected into a single memory vector $\mathbf{M}(t)$, which serves as the **Memory**. A standard Transformer decoder then refines $\mathbf{Q}$ by cross-attending to $\mathbf{M}(t)$, producing output features $\mathbf{F}_{\text{out}}(t)$ (Yu et al., 2021). These features are fed into four parallel MLP heads to predict the spatial coordinates $\hat{P}(t)$, the cell type logits $\hat{L}(t)$, the gene expressions $\hat{G}(t)$, and the point-wise existence logits $\hat{E}(t)$, with the latter being crucial for modeling population dynamics.

### 3.4. Multi-faceted Generative Loss Function

Our model is trained end-to-end by minimizing a multi-component loss $\mathcal{L}_{\text{total}}$. We use the existence logits $\hat{E}(t_i)$ to identify the valid predicted point set $\hat{S}_{\text{valid}}(t_i)$ and its corresponding ground truth set $S_{t_i}^{\text{GT}}$. The geometry loss $\mathcal{L}_{\text{geo}}$ combines both Chamfer distance (Fan et al., 2017) and Sinkhorn distance (Optimal Transport). The population loss $\mathcal{L}_{\text{exist}}$ is a Binary Cross-Entropy (BCE) on the existence logits. To supervise point-wise attributes, we define an attribute loss $\mathcal{L}_{\text{attr}} = \mathcal{L}_{\text{type}} + \mathcal{L}_{\text{gene}}$, where supervision is applied using the nearest-neighbor (NN) matching indices $\mathbf{A}_i$ from the Chamfer distance computation (Fan et al., 2017). To handle class imbalance, we apply a multi-class **Focal Loss** $\mathcal{L}_{\text{type}}$

(Lin et al., 2017) for cell types, and an $\mathcal{L}_2$ (MSE) loss $\mathcal{L}_{\text{gene}}$ for gene expression.

We further regularize the learned dynamics to encourage smooth and physically plausible temporal evolution. A standard VAE Kullback-Leibler divergence $\mathcal{L}_{\text{KL}} = \text{KL}(q(z(t_0)|c) \parallel p(z(t_0)))$ is applied to the initial state. A key component of our objective is the **Hybrid Trajectory Consistency Loss** ($\mathcal{L}_{\text{traj}}$), which ensures the learned velocity $f_\theta$ matches the empirical velocity $\mathbf{V}^{\text{emp}}$. We dynamically compute $\mathbf{V}^{\text{emp}}$ by finding nearest neighbors in a hybrid distance space $\mathbf{D}_{\text{hybrid}} = w_{\text{spatial}}\mathbf{Dis}_{\text{spatial}} + (1 - w_{\text{spatial}})\mathbf{Dis}_{\text{feature}}$(Vayer et al., 2020) , which combines normalized spatial and feature distances. $\mathcal{L}_{\text{traj}}$ is the MSE between $\mathbf{V}^{\text{emp}}$ and the equivalent model velocity $\mathbf{V}^{\text{model}}$. To further promote smooth temporal evolution, a **Latent Acceleration Regularization** ($\mathcal{L}_{\text{kin}}$) penalizes the second-order temporal derivative of $z(t)$ using finite differences:

$$\mathcal{L}_{\text{kin}} = \sum_i \left(\left\|\frac{z_{t_{i+1}} - z_{t_i}}{t_{i+1} - t_i}\right\|_2 - \left\|\frac{z_{t_i} - z_{t_{i-1}}}{t_i - t_{i-1}}\right\|_2\right)^2 \quad (8)$$

The final objective $\mathcal{L}_{\text{total}}$ is the weighted sum of all components:

$$\begin{aligned} \mathcal{L}_{\text{total}} = \sum_i \left(\mathcal{L}_{\text{geo}} + \mathcal{L}_{\text{exist}} + \mathcal{L}_{\text{attr}}\right)_i \\ + w_{\text{traj}}\mathcal{L}_{\text{traj}} + w_{\text{kin}}\mathcal{L}_{\text{kin}} + \beta\mathcal{L}_{\text{KL}} \end{aligned} \quad (9)$$

where $w$'s are scalar weights for each loss term and $\beta$ is the annealing coefficient for $\mathcal{L}_{\text{KL}}$. The specific settings of these parameters and additional formula details are provided in Appendix A.2.

## 4. Experiments

We comprehensively validate FlowCloud on multiple spatiotemporal datasets to demonstrate its advantages in generating continuous dynamics and unifying multimodal information, particularly when facing sparse, unpaired spatiotemporal transcriptomics data (Qiu et al., 2024; Klein et al., 2025).

### 4.1. Datasets and Evaluation Metrics

As summarized in Table 1, we employed four distinct dynamic datasets: 2D simulated data for evaluating complex spatiotemporal biological data, the Ambystoma brain dataset (Wei et al., 2022), the Drosophila embryo dataset (Wang et al., 2025), and the D-FAUST human motion dataset (Bogo et al., 2017) specifically for geometry fidelity evaluation. For the real-world data, we retained all cell types and 10 Spatially Variable Genes (SVGs). For the computationally intensive biological datasets, we focused on

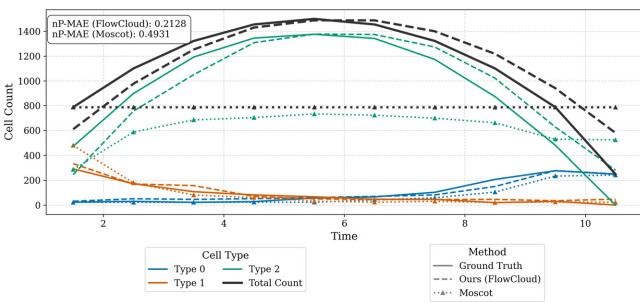

*Figure 5.* **Quantitative Comparison of Population Dynamics on the Simulation 2D Dataset.**

*Table 1.* **Dataset and Task Configuration.** Summary of the datasets, their purpose, total snapshots, the number of snapshots held out for testing, and the evaluation scope.

| Dataset | Purpose | Total Snaps | Test Snaps | Evaluation Scope |
|---|---|---|---|---|
| Simulation 2D | Validation | 20 | 10 | All Metrics |
| Ambystoma 2D | Real-world | 7 | 3 | All Metrics |
| Drosophila 3D | Real-world | 3 | 1 | All Metrics |
| Human Motion 3D | Extension | 100 | 50 | Geometry |

interpolation and attribute fidelity. For the 2D simulated dataset and D-FAUST human motion dataset, we present 10% and 2% temporal extrapolation results, respectively. Detailed information about the datasets and the mathematical formulation of the 2D simulated data are provided in Appendix B. We conduct experiments on all data using the same settings: 10,000 epochs and a learning rate of 1e-4. All experiments were conducted on a high-performance server equipped with four A100 GPUs (80G). Our multi-faceted evaluation strategy is designed to measure accuracy across geometric, attribute, and temporal dimensions. We adopt the following key metrics: **Geometry Fidelity:** Chamfer Distance (CD) (Achlioptas et al., 2018) ↓ for local matching and Sinkhorn Distance (OT) ↓ (Cuturi, 2013; Peyré & Cuturi, 2019; Achlioptas et al., 2018) for global distributional similarity. **Attribute Fidelity:** Cell Type Accuracy (CTA) ↑ and gene Spearman's Rank Correlation (SCC) ↑. Detailed formulas are provided in Appendix C.

### 4.2. Comparison with SOTA Baselines

We benchmarked FlowCloud against state-of-the-art (SOTA) methods across three categories: scene flow (Gu et al., 2019; Kittenplon et al., 2021), registration, and discrete Optimal Transport (OT). For scene flow methods incapable of producing continuous dynamics, we employed an autoregressive strategy to generate continuous spatiotemporal trajectories. Quantitative interpolation results are shown in Table 2. FlowCloud achieved substantial improvements in geometry fidelity (e.g., on the Human 3D dataset, the CD was reduced by 43.0% compared to the best baseline, SPCMNet). Quan-

titative results for attribute interpolation, including CTA and SCC, are summarized in Table 3. We further visualized manifold reconstructions of the 2D simulated data and inferred cell types for the Drosophila embryo across different time points in Figure 6 and Figure 7, respectively. The accuracy of predicted population dynamics is demonstrated in Figure 5 (see Appendix D.1 for details). Overall, these results demonstrate that FlowCloud consistently outperforms competing methods in both geometric reconstruction accuracy and attribute fidelity. Visualization results for the axolotl brain data and human motion data, together with further analyses of group-level dynamics, are provided in Appendix D.

We evaluate the stability of the learned continuous velocity field by testing generalization to unseen future time points ($T_{future}$). Specifically, we assess extrapolation performance with 10% and 2% future horizons on the 2D simulated dataset and the D-FAUST human motion dataset (Chen et al., 2018). Quantitative metrics for 5 repeated extrapolations under the same model are reported in Table 4, and the extrapolated results are visualized in Figure 6 at the time points marked with an asterisk (*). These results indicate that FlowCloud is able to maintain coherent and stable dynamics when extrapolating beyond a small observed temporal range.

### 4.3. Ablation Study on FlowCloud Architecture

We conducted an ablation study on the 2D simulated dataset (Table 5). The results validate the necessity of our complete design. Specifically, removing the Neural ODE (Variant C) leads to a severe degradation in extrapolation performance (extrapolation CD surged to 0.5517), highlighting the critical role of the continuous velocity field (Dupont et al., 2019; Kidger et al., 2020; Grathwohl et al., 2019). Similarly, replacing the Point Swin Transformer with PointNet++ (Variant A) also causes a significant performance drop (CTA decreased from 0.898 to 0.812), demonstrating its importance in extracting features from sparse topologies. Furthermore, removing the global context (Variant B), the trajectory loss (Variant D), or the attention aggregator (Variant E) all led to varying degrees of performance degradation, confirming the contribution of each component in our end-to-end design.

### 4.4. Case Study: Ambystoma Injury and Regeneration Dynamics

We selected the Ambystoma injury dataset for an in-depth qualitative case study. This dataset comprises 7 non-contiguous temporal snapshots and provides a challenging real-world benchmark for evaluating continuous spatiotemporal modeling in biological development and tissue regeneration.

*Table 2.* Quantitative comparison of **Geometrical Fidelity** on all datasets. We report Mean ± SD across all test time points, except for Drosophila 3D (single test snapshot), which uses 5 random-seeded runs. (↓) Lower is better. The best result for each metric is highlighted in **bold**.The '0.0000' denotes a non-zero value below the displayed precision (applies to all tables).

| Method | Simulation 2D | | Ambystoma 2D | | Drosophila 3D | | Human 3D | |
|---|---|---|---|---|---|---|---|---|
| | OT (↓) | CD (↓) | OT (↓) | CD (↓) | OT (↓) | CD (↓) | OT (↓) | CD (↓) |
| FlowNet3D (CVPR (Liu et al., 2019a)) | 0.8417 ± 0.5562 | 1.7305 ± 0.8490 | 0.8870 ± 0.5437 | 0.9018 ± 0.5709 | 0.3571 ± 0.0114 | 0.7505 ± 0.0093 | 0.4778 ± 0.4651 | 0.5564 ± 0.1975 |
| DifFlow3D (CVPR (Liu et al., 2024)) | 0.5011 ± 0.3298 | 1.0679 ± 0.5188 | **0.1748 ± 0.0311** | 0.2780 ± 0.0349 | 0.2488 ± 0.0121 | 0.5370 ± 0.0112 | 0.0051 ± 0.0139 | 0.1231 ± 0.1032 |
| Meteornet (ICCV (Liu et al., 2019b)) | 0.7099 ± 0.4521 | 0.7597 ± 0.3614 | 0.2881 ± 0.1462 | 0.2454 ± 0.0536 | **0.2021 ± 0.0095** | 0.5360 ± 0.0109 | 0.0020 ± 0.0012 | 0.0799 ± 0.0144 |
| SPCMNet (IJCV (He et al., 2022)) | 0.0870 ± 0.0930 | 0.2312 ± 0.0360 | 1.1441 ± 0.9655 | 0.2330 ± 0.0301 | 0.3509 ± 0.0118 | 0.6481 ± 0.0110 | 0.0021 ± 0.0006 | 0.0470 ± 0.0042 |
| VoteFlow (CVPR (Lin et al., 2025b)) | 0.7645 ± 0.4625 | 1.4353 ± 0.6858 | 0.2760 ± 0.0642 | 0.5771 ± 0.0689 | 0.4220 ± 0.0092 | 1.2435 ± 0.0123 | 0.0004 ± 0.0000 | 0.0474 ± 0.0034 |
| Spateo (Cell (Qiu et al., 2024)) | 0.0179 ± 0.0284 | 1.6060 ± 0.0313 | 0.3610 ± 0.1753 | 0.2410 ± 0.0326 | 0.3328 ± 0.0013 | 0.5041 ± 0.0031 | - | - |
| Moscot (Nature (Klein et al., 2025)) | 0.5178 ± 0.2199 | 1.0175 ± 0.2140 | 0.9158 ± 0.5897 | 0.9088 ± 0.4819 | 0.2540 ± 0.0025 | 0.4918 ± 0.0015 | - | - |
| **FlowCloud (Ours)** | **0.0147 ± 0.0114** | **0.1369 ± 0.0226** | 0.2868 ± 0.0916 | **0.2261 ± 0.0647** | 0.2238 ± 0.0040 | **0.4772 ± 0.0094** | **0.0003 ± 0.0000** | **0.0268 ± 0.0027** |

*Table 3.* Quantitative comparison of **Attribute Fidelity** on all datasets. (↑) Higher is better. The best result for each metric is highlighted in **bold**. The Human 3D dataset is omitted as it does not contain attribute (cell type/gene) labels.

| Method | Simulation 2D | | Ambystoma 2D | | Drosophila 3D | |
|---|---|---|---|---|---|---|
| | CTA (↑) | SCC (↑) | CTA (↑) | SCC (↑) | CTA (↑) | SCC (↑) |
| FlowNet3D (CVPR (Liu et al., 2019a)) | 0.2849 ± 0.1753 | 0.1981 ± 0.0829 | 0.1451 ± 0.0301 | 0.0685 ± 0.0162 | 0.0253 ± 0.0115 | 0.1048 ± 0.0092 |
| DifFlow3D (CVPR (Liu et al., 2024)) | 0.3188 ± 0.1945 | 0.2959 ± 0.0590 | 0.2008 ± 0.0447 | 0.1214 ± 0.0228 | 0.0549 ± 0.0123 | 0.0897 ± 0.0114 |
| Meteornet (ICCV (Liu et al., 2019b)) | 0.6215 ± 0.2112 | 0.3138 ± 0.1415 | 0.1087 ± 0.0837 | 0.0953 ± 0.0458 | 0.0366 ± 0.0091 | 0.0035 ± 0.0108 |
| SPCMNet (IJCV (He et al., 2022)) | 0.2857 ± 0.1061 | 0.3860 ± 0.0982 | 0.1201 ± 0.0045 | 0.0188 ± 0.0109 | 0.0295 ± 0.0113 | 0.0291 ± 0.0121 |
| VoteFlow (CVPR (Lin et al., 2025b)) | 0.2417 ± 0.2719 | 0.0250 ± 0.0133 | 0.0092 ± 0.0009 | 0.0088 ± 0.0067 | 0.0026 ± 0.0094 | 0.0121 ± 0.0112 |
| Spateo (Cell (Qiu et al., 2024)) | 0.5719 ± 0.1748 | 0.5030 ± 0.0727 | 0.2095 ± 0.0504 | 0.1270 ± 0.0040 | 0.0193 ± 0.0110 | 0.0010 ± 0.0003 |
| Moscot (Nature (Klein et al., 2025)) | 0.5521 ± 0.1683 | 0.3187 ± 0.0788 | 0.1064 ± 0.0132 | 0.0369 ± 0.0134 | 0.0442 ± 0.0009 | 0.0331 ± 0.0002 |
| **FlowCloud (Ours)** | **0.8970 ± 0.1632** | **0.6273 ± 0.0806** | **0.2521 ± 0.0701** | **0.1960 ± 0.0307** | **0.1086 ± 0.032** | **0.1325 ± 0.0044** |

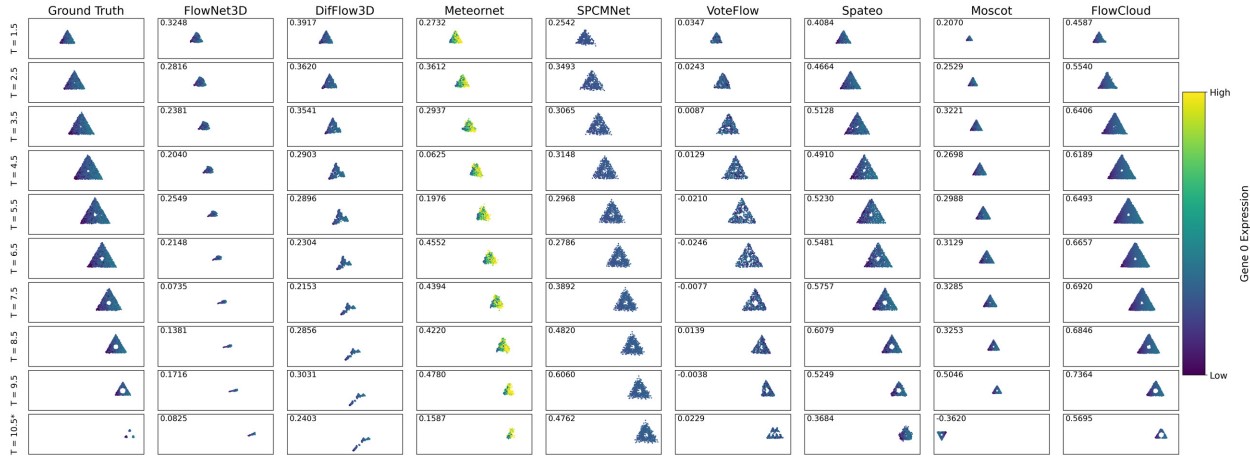

*Figure 6.* Qualitative Comparison of Spatiotemporal Gene Expression Trajectories on the Simulation 2D Dataset, evaluated by Gene Expression (SCC ↑).

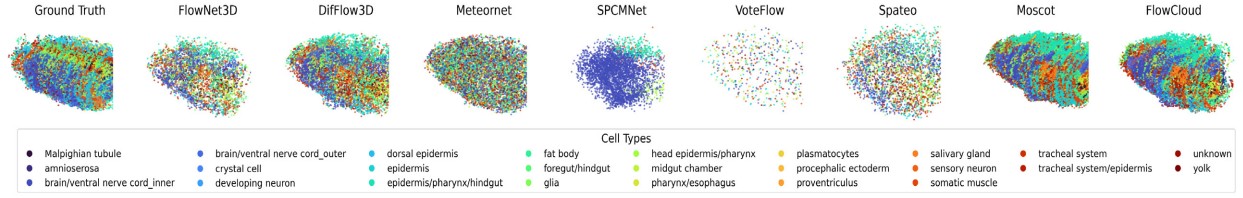

*Figure 7.* Qualitative Comparison of Cell-Type Reconstructions on the Drosophila 3D Dataset, evaluated by Cell Type Accuracy (CTA ↑).

### 4.4.1. SPATIOTEMPORAL RECONSTRUCTION AND CELL TRAJECTORY PREDICTION

FlowCloud leverages its continuous latent dynamic model $z(t)$ to successfully interpolate and reconstruct the continuous, full spatiotemporal dynamics of cells and genes from the 7 sparse observational time points. As shown in Figure 8, FlowCloud achieves coherent spatiotemporal reconstructions of spatial morphology, discrete cell types, gene expression dynamics, and long-range cellular trajectories of CMPN (Cholinergic, Monoaminergic, and Peptidergic Neuron).

*Table 4.* Quantitative comparison of Extrapolation Performance. We report Mean ± Standard Deviation on unseen future time points (extrapolation). (↓) Lower is better for OT/CD; (↑) Higher is better for CTA/SCC. The best result for each metric is highlighted in bold.

| Method | Simulation 2D (T=10.5) | | | | Human 3D (T=100.0) | |
|---|---|---|---|---|---|---|
| | OT (↓) | CD (↓) | CTA (↑) | SCC (↑) | OT (↓) | CD (↓) |
| FlowNet3D | $0.3615 \pm 0.0027$ | $1.1310 \pm 0.0141$ | $0.0439 \pm 0.0011$ | $0.0822 \pm 0.0031$ | $1.6922 \pm 0.0231$ | $0.5875 \pm 0.0053$ |
| DifFlow3D | $1.0271 \pm 0.0192$ | $1.2701 \pm 0.0211$ | $0.0243 \pm 0.0010$ | $0.2408 \pm 0.0042$ | $0.0004 \pm 0.0000$ | $0.0496 \pm 0.0023$ |
| Meteornet | $0.3288 \pm 0.0031$ | $0.7819 \pm 0.0078$ | $0.1201 \pm 0.0024$ | $0.1592 \pm 0.0033$ | $0.0013 \pm 0.0003$ | $0.0719 \pm 0.0011$ |
| SPCMNet | $0.3281 \pm 0.0029$ | $0.2578 \pm 0.0041$ | $0.0002 \pm 0.0001$ | $0.4453 \pm 0.0048$ | $0.0031 \pm 0.0005$ | $0.0596 \pm 0.0008$ |
| VoteFlow | $0.1010 \pm 0.0072$ | $0.1512 \pm 0.031$ | $0.4056 \pm 0.0018$ | $0.0229 \pm 0.0025$ | $0.0003 \pm 0.0000$ | $0.0496 \pm 0.0001$ |
| Spateo | $0.0980 \pm 0.0016$ | $0.1410 \pm 0.0021$ | $0.1369 \pm 0.0031$ | $0.3681 \pm 0.0047$ | - | - |
| Moscot | $0.2961 \pm 0.0022$ | $0.8584 \pm 0.0088$ | $0.2373 \pm 0.0041$ | $0.3619 \pm 0.0051$ | - | - |
| **FlowCloud (Ours)** | $\mathbf{0.0449 \pm 0.0009}$ | $\mathbf{0.1119 \pm 0.0014}$ | $\mathbf{0.4581 \pm 0.0049}$ | $\mathbf{0.5690 \pm 0.0062}$ | $\mathbf{0.0003 \pm 0.0000}$ | $\mathbf{0.0225 \pm 0.0005}$ |

*Table 5.* **Ablation study on the Simulation 2D dataset.** We validate the effectiveness of each component in FlowCloud. Metrics reported are Chamfer Distance (CD) for both interpolation ($t \in [t_1, t_N]$) and extrapolation ($t > t_N$), and Cell Type Accuracy (CTA). **Extrap. Failure** denotes a severe degradation in topology preservation.

| Model Variant | Key Components | | | | Performance Metrics (Sim 2D) | | |
|---|---|---|---|---|---|---|---|
| | Encoder | Aggregator | Dynamics | Loss | Interp. CD (↓) | Extrap. CD (↓) | CTA (↑) |
| *A. w/o Swin Transformer* | PointNet++ | Seq. Trans | Neural ODE | Full | 0.0380 | 0.0912 | 0.812 |
| *B. w/o Global Context* | Point Swin | Recursive | Neural ODE | Full | 0.0259 | 0.3105 | 0.763 |
| *C. w/o Neural ODE* | Point Swin | Seq. Trans | Discrete | Full | 0.0211 | 0.5517 | 0.790 |
| *D. w/o Trajectory Loss* | Point Swin | Seq. Trans | Neural ODE | w/o $\mathcal{L}_{traj}$ | 0.0187 | 0.0691 | 0.853 |
| *E. w/o Attention Agg.* | Point Swin | Mean Pool | Neural ODE | Full | 0.0196 | 0.0840 | 0.860 |
| **FlowCloud (Full)** | **Point Swin** | **Seq. Trans** | **Neural ODE** | **Full** | **0.0147** | **0.0449** | **0.898** |

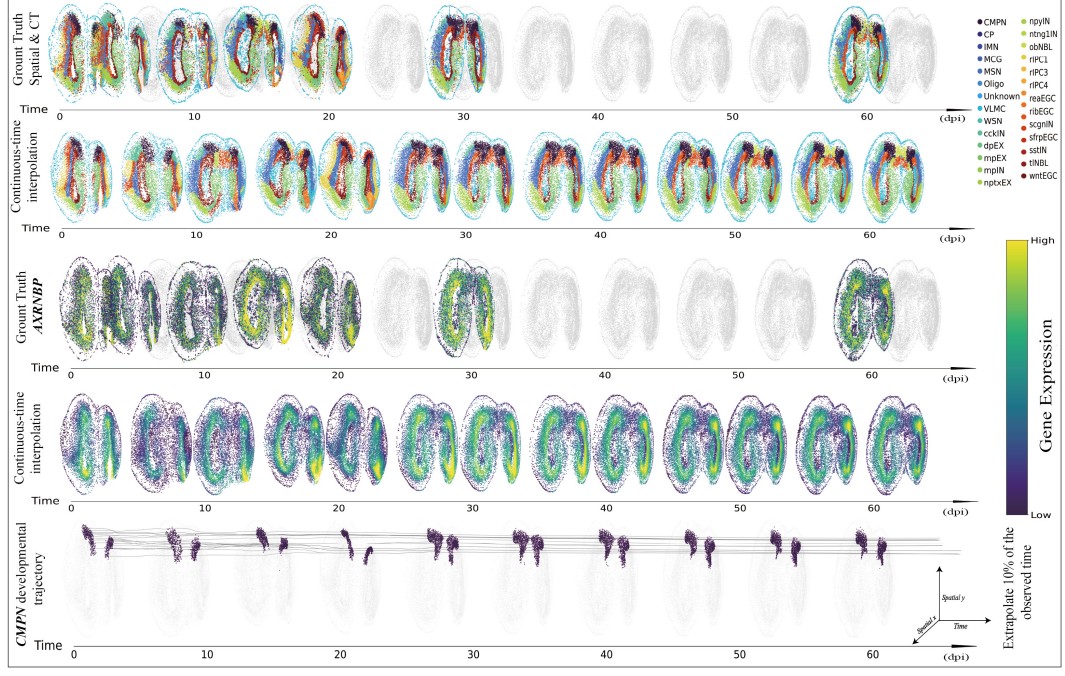

*Figure 8.* Continuous Reconstruction of Ambystoma Injury–Regeneration Dynamics Across Space, Time, and Gene Expression

### 4.4.2. INSTANTANEOUS CELL DEVELOPMENT VELOCITY FIELD

We visualized the cell velocity vector field predicted by FlowCloud at 5 days post-injury (dpi). As shown in Figure 9, the inferred vector field reveals coordinated cell migration from the surrounding injury region toward the wound center.

In particular, stem cells such as realEGC exhibit directed movement toward the injury site, consistent with the regenerative process reported by (Qiu et al., 2022). This result maps FlowCloud's learned continuous latent velocity field $\frac{dz}{dt} = f_\theta(z(t), t)$ onto physical space, demonstrating the model's ability to capture both local and global instanta-

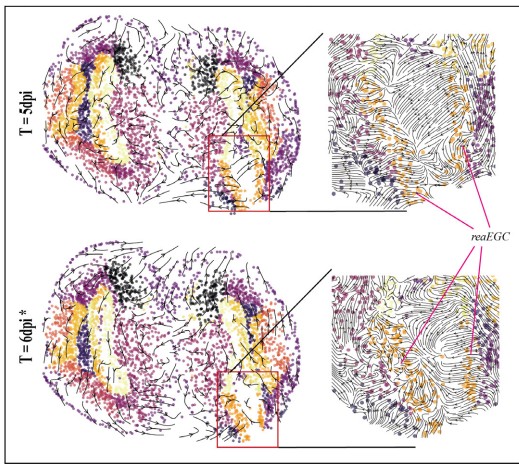

*Figure 9.* Instantaneous Cell Velocity Field During Ambystoma Regeneration (T=6dpi* represents an unobserved slice).

neous motion patterns. Furthermore, we use FlowCloud to predict cell positions and the velocity vector field at an unobserved time point (6 dpi), further supporting the model's capability for continuous spatiotemporal prediction (Bergen et al., 2020; Lange et al., 2022).

## 5. Conclusion

We introduce FlowCloud, a variational Neural ODE framework designed to reconstruct continuous spatiotemporal dynamics from sparse, unpaired point cloud snapshots. Unlike paired scene flow or discrete Optimal Transport methods, FlowCloud learns a unified continuous velocity field. By defining a global initial state $z(t_0)$ via a Point Swin Transformer and sequence aggregator, it avoids recursive cumulative errors. A composite loss (Sinkhorn, Chamfer, trajectory regularization) ensures physical plausibility. FlowCloud significantly outperforms SOTA methods on simulated, human motion, and transcriptomics benchmarks, providing a unified end-to-end solution for continuous temporal reconstruction from unstructured observations (Grathwohl et al., 2019; Yang et al., 2019).

## 6. Discussion

Our results demonstrate that initializing continuous models from globally aggregated context is more robust than frame-by-frame propagation for sparse, unpaired data. In the axolotl regeneration case (Figs. 8 and 9), FlowCloud learned meaningful migration velocity fields, suggesting its potential applicability to developmental biology studies. However, limitations remain. While short-term extrapolation is effective, Neural ODEs risk dynamics drift in long-term predictions (Linot et al., 2023; Rodriguez et al., 2022). In addition, the current architecture may encounter computa-

tional bottlenecks when scaling to extremely large numbers of snapshots or million-scale point clouds (Wu et al., 2022; Qian et al., 2022). Finally, although population changes are modeled through an "existence predictor", complex biological processes would require richer and more structured modeling (Chizat et al., 2018). Future work will explore the integration of more sophisticated biological priors to further improve population dynamics modeling (Forrow & Schiebinger, 2021; Bergen et al., 2020).

## Limitations

Despite its efficacy, FlowCloud has several limitations. First, its soft alignment struggles with extreme rigid rotations ($> 30°$), currently requiring preprocessing (e.g., PASTE), though future work could embed explicit rigid transformations for alignment-free learning. Second, the framework is inherently a smoother designed for retrospective historical reconstruction rather than a real-time causal forecaster. Finally, to prioritize multimodal scalability, we use a simplified existence predictor instead of Unbalanced Optimal Transport (e.g., stVCR, TIGON) to model cell birth and death. Integrating physically grounded mass conservation remains a promising future direction.

## Acknowledgements

This research was funded by the Lei Jun Foundation Limited and the Fundamental Research Funds for the Central Universities (Grant No. 2042025kf0038).

## Impact Statement

FlowCloud provides a robust generative framework for reconstructing continuous biological and motion dynamics from sparse, unpaired observations. By unifying geometry, gene expression, and population shifts, it advances precision medicine and developmental biology. This capability enables deeper insights into cellular differentiation and morphogenesis without requiring dense temporal measurements. The source code for this project is available at https://github.com/yinboliu-git/FlowCloud.

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

# A. Model Details

## A.1. Model Architecture Details

We provide the detailed layer-by-layer specification of the FlowCloud model architecture in Table 6. The dimensions shown correspond to the Ambystoma 2D dataset ($D_{spatial} = 2$, $N_{types} = 27$, $N_{genes} = 10$). The total input feature dimension is $D_{in} = 2 + 27 + 10 = 39$.

*Table 6.* FlowCloud Architecture. Dimensions are for the Ambystoma 2D dataset.

| Component | Module / Layer | Specification | Output Dimension |
|---|---|---|---|
| **Point Encoder** | **Input Features** | **(Coords + Types + Genes)** | $(T, N, 39)$ |
| | `block1`/`hfl1` | `npoint=1024, k=32` `Linear(39) -> 64` | $(T, 1024, 64)$ |
| | `block2`/`hfl2` | `npoint=256, k=32` `Linear(66) -> 128` | $(T, 256, 128)$ |
| | `block2`/`hfl1` | `npoint=64, k=32` `Linear(130) -> 256` | $(T, 64, 256)$ |
| | `block2`/`hfl2` | `npoint=16, k=16` `Linear(258) -> 512` | $(T, 16, 512)$ |
| | **Global Max Pooling** | **Pools over 16 points** | $(T, 512)$ |
| | `output_mlp` | `Linear(512) -> ReLU ->` `Linear(256)` | $(T, 256)$ |
| | `(Unsqueeze)` | **Adds batch dimension for Transformer** | $(T, 1, 256)$ |
| **Aggregate Enc.** | `time_encoder` | `TimeEncoding` | $(T, 1, 256)$ |
| | `sequence_encoder` | `TransformerEncoder (3 layers)` | $(T, 1, 256)$ |
| | **Global Mean Pooling** | **Pools over $T$ time steps** | $(1, 256)$ |
| **Latent Init.** | `latent_to_z0` | `Linear(256) -> 512` | $(1, 512)$ |
| | **Split & Reparameterize** | $(z_\mu(128), z_{\log \sigma^2}(128))$ **& Squeeze** | $(256)$ |
| **Dynamics Func.** | `(Input)` | $z$ **and** $t$ | $(256)$ and $(1)$ |
| | `time_encoder` | `Linear(1) -> ReLU -> Linear(32)` | $(32)$ |
| | `(Concatenate)` | **Concatenate $z$ and time features** | $(288)$ |
| | `net` | `MLP(288 → 256 → 256 → 256) +` `Tanh` | $(256)$ |
| | **ODE Solver** | **`odeint(func, z0, t_eval)`** | $(T_{\text{eval}}, 256)$ |
| **Decoder** | `queries` | `nn.Parameter` | $(N_{\text{pad}}, 256)$ |
| | `latent_proj` | `Linear(256) → 256` | $(T_{\text{eval}}, 256)$ |
| | `(Prepare Inputs)` | **`memory` and `tgt`** | $(T_{\text{eval}}, 1, 256)$ & $(T_{\text{eval}}, N_{\text{pad}}, 256)$ |
| | `transformer_decoder` | `TransformerDecoder (2 layers)` | $(T_{\text{eval}}, N_{\text{pad}}, 256)$ |
| | `shape_head` | `MLP(256 → 128 → 2)` | $(T_{\text{eval}}, N_{\text{pad}}, 2)$ |
| | `type_head` | `MLP(256 → 128 → 27)` | $(T_{\text{eval}}, N_{\text{pad}}, 27)$ |
| | `gene_head` | `MLP(256 → 128 → 10)` | $(T_{\text{eval}}, N_{\text{pad}}, 10)$ |
| | `existence_head` | `MLP(256 → 128 → 1)` | $(T_{\text{eval}}, N_{\text{pad}}, 1)$ |

## A.2. Loss Function Details

### A.2.1. TRAJECTORY CONSISTENCY LOSS

Trajectory Consistency Loss ($\mathcal{L}_{traj}$) is crucial for non-paired data, as it enforces that the model's learned velocity field ($\mathbf{V}^{model}$) aligns with an empirical velocity field ($\mathbf{V}^{emp}$) derived from the data.

The key innovation is that the Nearest Neighbor (NN) matching, used to compute $\mathbf{V}^{emp}$, is not based on spatial distance alone. Instead, it uses a hybrid distance metric that combines both normalized spatial and normalized feature (gene) distances. The spatial and feature distances are the squared Euclidean distances of Z-score normalized components:

$$Dis_{spatial}(p_i, p_j) = \left\| \hat{P}_{A,i} - \hat{P}_{B,j} \right\|_2^2$$

$$Dis_{feature}(f_i, f_j) = \left\| \hat{F}_{A,i} - \hat{F}_{B,j} \right\|_2^2$$

The final hybrid distance ($D_{hybrid}$) used for matching is a weighted sum:

$$D_{hybrid} = w_{spatial} \cdot Dis_{spatial} + (1 - w_{spatial}) \cdot Dis_{feature}$$

In our implementation, the spatial weight $w_{spatial}$ = 0.9 (figure 9). The loss is then computed for each pair of time points $t_i \to t_{i+1}$:

1. **Empirical Velocity ($\mathbf{V}^{emp}$):** For each point $p_i \in A$, find its nearest neighbor $p_j \in B$ using the $D_{hybrid}$ metric. Compute the empirical velocity $v_{emp} = (\hat{p}_j - \hat{p}_i)/\Delta t$.

2. **Model Velocity ($\mathbf{V}^{model}$):** Perform the same hybrid NN matching and velocity calculation for the model's normalized reconstructions ($\hat{A}_{recon}, \hat{B}_{recon}$).

3. **Loss Computation:** The $\mathbf{V}^{emp}$ and $\mathbf{V}^{model}$ fields are aligned (also using $D_{hybrid}$ for a bidirectional Chamfer-style alignment between $A$ and $\hat{A}_{recon}$), and the final loss is the $\mathcal{L}_2$ (MSE) between them.

### A.2.2. LOSS FUNCTION WEIGHTS

The weights ($w_k, \beta$) for the final objective $L_{total}$ are defined in our training script as specified in Table 7.

*Table 7.* Hyperparameters for the composite loss function.

| Loss Term | Code Variable | Symbol | Value |
|---|---|---|---|
| $L_{geo}$ (Chamfer) | `alpha_recon_loss` | $\alpha_{geo}$ | 5.0 |
| $L_{geo}$ (Sinkhorn) | `alpha_recon_loss` | $\alpha_{geo}$ | 5.0 |
| $L_{type}$ | `gamma_type_loss` | $\gamma_{type}$ | 1.0 |
| $L_{gene}$ | `gamma_gene_loss` | $\gamma_{gene}$ | 1.0 |
| $L_{exist}$ | `gamma_existence_loss` | $\gamma_{exist}$ | 1.0 |
| $L_{traj}$ | `gamma_trajectory_loss` | $w_{traj}$ | 1.0 |
| $L_{kin}$ | `lambda_kinetic_reg` | $w_{kin}$ | 1e-7 |
| $L_{KL}$ | `current_beta` | $\beta$ | 0.0 → 1.0 (Annealed) (Kingma & Welling, 2014) |
| Focal Loss $\gamma$ | `focal_loss_gamma` | $\gamma$ | 2.0 (Lin et al., 2017) |
| Hybrid Dist. Spatial Weight | `w_spatial` | $w_{spatial}$ | 0.9 |

### A.3. Training and Implementation Details

- **Optimizer:** We use `torch.optim.Adam`.

- **Learning Rate Scheduler:** We use `StepLR` with `step_size=1000` and `gamma=0.8`.

- **Weight Decay:** 1e-7.

- **Batching:** Our model is trained using full-batch, where the sequence encoder $\mathcal{E}_{seq}$ processes all $N$ observed snapshots in a single step.

- **Hardware:** As mentioned, training was performed on 4x A100 (80G) GPUs.

### A.4. Hyperparameter and Running Time Analysis

We conduct a detailed analysis of computational performance and parameter sensitivity, examining the impact of critical factors such as model complexity, learning rates, and sampling density. The comprehensive results are summarized in Table 8 and 9, with detailed sensitivity curves presented in Figures 10, 11, and 12. It is important to note that the running times reported in Table 9 serve solely as a reference. For most baseline models, the total training time is estimated based on the average time of running 5 epochs, while for Spateo and Moscot, the reported times are empirical estimates. Overall, FlowCloud does not exhibit a computational speed advantage over lightweight baselines, as it inherently trades training efficiency for the capacity to perform unified continuous spatiotemporal and multi-attribute modeling.

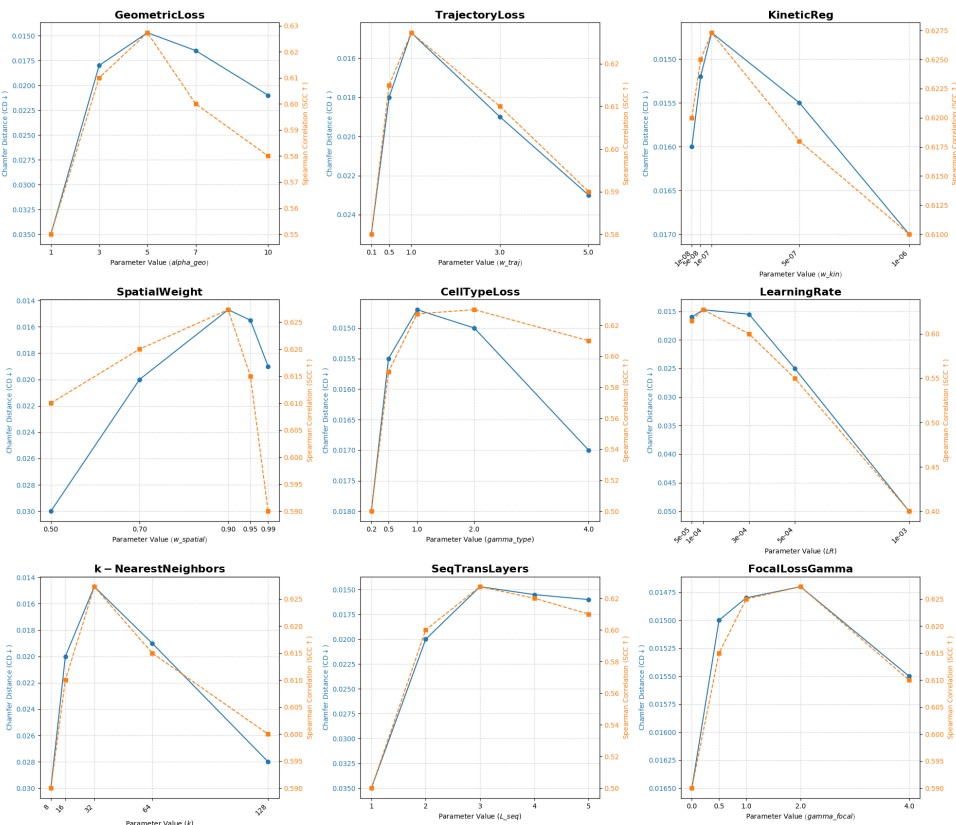

*Figure 10.* Estimated **Hyperparameter Sensitivity Analysis** on the **Simulation 2D** Dataset. The $3 \times 3$ subplot array uses dual Y-axes to display the trade-off between Chamfer Distance (CD $\downarrow$, Left Axis) and Spearman Correlation (SCC $\uparrow$, Right Axis) across 9 critical parameters (including loss weights, learning rate, and encoder architecture).

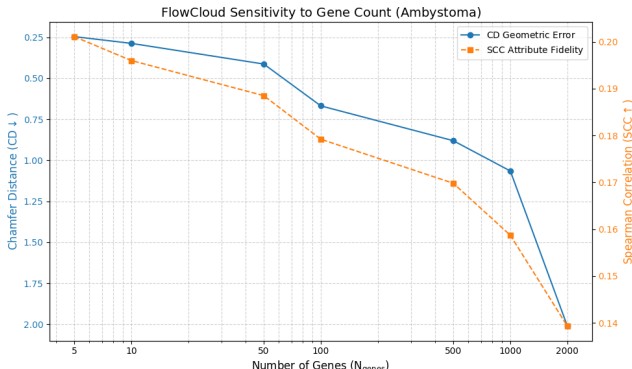

*Figure 11.* Estimated Model Sensitivity to the **Number of Genes ($N_{genes}$)** on the **Ambystoma 2D** Dataset. The plot uses a dual Y-axis and an X-axis with custom ticks to demonstrate the strong increase in geometric error (CD) as $N_{genes}$ increases, while attribute fidelity (SCC) remains relatively stable. $N_{genes} = 10$ is the anchor point used in the main experiments.

## A.5. Time Complexity Analysis

We analyze the time complexity $\mathcal{O}_{\text{FlowCloud}}$ based on the dominant operations during the model's forward pass, where $T$ is the number of input snapshots, $N$ is the average number of points per snapshot, $M$ is the maximum point capacity of the decoder, and $D$ is the latent dimension.

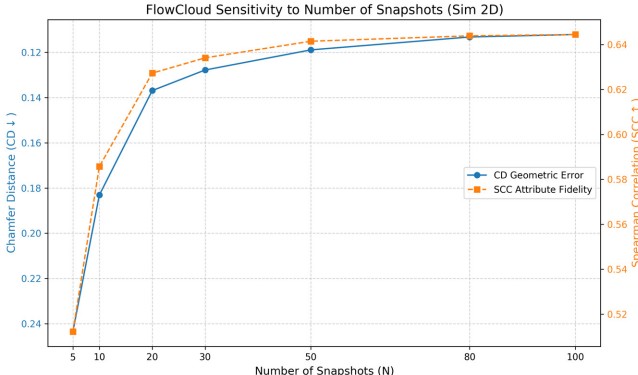

*Figure 12.* Estimated Model Sensitivity to the **Number of Snapshots (N)** on the **Simulation 2D** Dataset. The plot shows that performance improves with higher sampling density ($N$), exhibiting clear diminishing returns after the anchor point ($N = 20$).

*Table 8.* Estimated Hyperparameter Search Ranges and Optimal Values for Baseline Models.

| Model | Hyperparameter | Search Range | Sim Optimal | Amby Optimal | Dros Optimal | Human Optimal |
|---|---|---|---|---|---|---|
| FlowNet3D (Liu et al., 2019a) | learning rate | $[5.0 \times 10^{-5}, 3.0 \times 10^{-4}]$ | $1.0 \times 10^{-4}$ | $5.0 \times 10^{-5}$ | $5.0 \times 10^{-5}$ | $1.0 \times 10^{-4}$ |
| | spatial_weight | $[0.5, 0.9]$ | 0.7 | 0.8 | 0.8 | 0.6 |
| | iters/refine | $[500, 1500]$ | 1000 | 1500 | 1500 | 1000 |
| DifFlow3D (Liu et al., 2024) | learning rate | $[5.0 \times 10^{-5}, 3.0 \times 10^{-4}]$ | $1.0 \times 10^{-4}$ | $5.0 \times 10^{-5}$ | $5.0 \times 10^{-5}$ | $1.0 \times 10^{-4}$ |
| | weight decay | $[1.0 \times 10^{-6}, 1.0 \times 10^{-4}]$ | $1.0 \times 10^{-5}$ | $5.0 \times 10^{-5}$ | $5.0 \times 10^{-5}$ | $1.0 \times 10^{-5}$ |
| | iters | $[500, 1500]$ | 1000 | 1500 | 1500 | 1000 |
| Meteornet (Liu et al., 2019b) | learning rate | $[5.0 \times 10^{-4}, 3.0 \times 10^{-3}]$ | $1.0 \times 10^{-3}$ | $8.0 \times 10^{-4}$ | $8.0 \times 10^{-4}$ | $1.0 \times 10^{-3}$ |
| | hidden dim | $[64, 256]$ | 128 | 192 | 192 | 128 |
| | Type Loss Weight | $[0.5, 2.0]$ | 1.0 | 1.5 | 1.5 | N/A |
| | Gene Loss Weight | $[0.5, 2.0]$ | 1.0 | 1.5 | 1.5 | N/A |
| SPCM (He et al., 2022) | learning rate | $[5.0 \times 10^{-4}, 3.0 \times 10^{-3}]$ | $1.0 \times 10^{-3}$ | $8.0 \times 10^{-4}$ | $8.0 \times 10^{-4}$ | $1.0 \times 10^{-3}$ |
| | weight decay | $[1.0 \times 10^{-6}, 1.0 \times 10^{-4}]$ | $1.0 \times 10^{-5}$ | $5.0 \times 10^{-5}$ | $5.0 \times 10^{-5}$ | $1.0 \times 10^{-5}$ |
| | iters | $[5000, 20000]$ | 10000 | 15000 | 15000 | 8000 |
| | Type Loss Weight | $[0.5, 2.0]$ | 1.0 | 1.5 | 1.5 | N/A |
| | Gene Loss Weight | $[0.5, 2.0]$ | 1.0 | 1.5 | 1.5 | N/A |
| VoteFlow (Lin et al., 2025b) | learning rate | $[5.0 \times 10^{-5}, 3.0 \times 10^{-4}]$ | $1.0 \times 10^{-4}$ | $5.0 \times 10^{-5}$ | $5.0 \times 10^{-5}$ | $1.0 \times 10^{-4}$ |
| | n_iters | $[500, 1500]$ | 1000 | 1500 | 1500 | 1000 |
| Moscot (Klein et al., 2025) | $\alpha$ | $[0.1, 1.0]$ | 0.5 | 0.6 | 0.6 | N/A |
| | rank | $[500, 1500]$ | 1000 | 1200 | 1200 | N/A |
| | Epsilon | $[1.0 \times 10^{-4}, 1.0 \times 10^{-2}]$ | $1.0 \times 10^{-3}$ | $5.0 \times 10^{-4}$ | $5.0 \times 10^{-4}$ | N/A |
| Spateo (Qiu et al., 2024) | $\alpha$ | $[0.1, 1.0]$ | 0.5 | 0.6 | 0.6 | N/A |

1. **Point Encoder (Swin Hierarchical):** The encoder operates per snapshot. Due to the hierarchical nature (FPS and grouping), the computational cost is approximately linear with respect to the input point count: $\mathcal{O}(T \cdot N)$ (Qi et al., 2017).

2. **Sequence Aggregator (Transformer):** The self-attention mechanism in the sequence Transformer scales quadratically with the sequence length (number of snapshots, $T$): $\mathcal{O}(T^2 \cdot D^2)$ (Vaswani et al., 2017).

3. **Decoder / ODE Integration:** The Neural ODE integration complexity is proportional to the number of evaluation steps ($T_{\text{eval}}$) and the complexity of the latent function $\mathcal{O}(D^2)$ (Chen et al., 2018). The attention-based decoder scales linearly with the maximum output points $M$ (Carion et al., 2020).

The overall complexity is thus:

$$\mathcal{O}_{\text{FlowCloud}} \approx \mathcal{O}\left(T \cdot N \cdot D_{\text{in}} + T^2 \cdot D^2 + M \cdot T_{\text{eval}} \cdot D^2\right)$$

This structure shows that for datasets with a very large number of points ($N \gg T$), the bottleneck is the Point Encoder; conversely, for long sequences ($T \gg N$), the bottleneck shifts to the temporal $\mathcal{O}(T^2)$ self-attention and the subsequent ODE integration.

*Table 9.* Comparison of Total Training Time (in Hours) across different datasets.

| Method | Simulation 2D | Ambystoma 2D | Drosophila 3D | Human 3D |
|---|---|---|---|---|
| FlowNet3D (Liu et al., 2019a) | 2.1374 | 5.1841 | 4.9036 | 5.7628 |
| DiffFlow3D (Liu et al., 2024) | 4.4318 | 0.4376 | 0.4872 | 2.6239 |
| Meteornet (Liu et al., 2019b) | 0.0758 | 0.1094 | 0.1417 | 0.2963 |
| SPCMNET (He et al., 2022) | 1.7421 | 0.9618 | 1.5274 | 5.3846 |
| VoteFlow (Lin et al., 2025b) | 4.4185 | 1.6732 | 1.4879 | 3.7024 |
| Spateo (Qiu et al., 2024) | $\approx 1.00$ | $> 10.00$ | $> 10.00$ | - |
| Moscot (Klein et al., 2025) | $\approx 2.00$ | $> 10.00$ | $> 10.00$ | - |
| **FlowCloud (Ours)** | **2.3147** | **3.8672** | **10.8419** | **14.7365** |

## B. Dataset Details

As promised in the main paper, we provide details on the simulated and real-world datasets used. **Data Splitting Strategy:** For all dynamic datasets, we adopt a temporal skipping approach where every $k$-th snapshot is sampled for the test set, ensuring the training and testing sets are temporally non-contiguous. For example, in the Simulation 2D dataset (20 total snapshots, 10 test snapshots ), we use a skipping factor of $k = 2$. The number of points contained in the datasets is shown in Table 10.

*Table 10.* Overview of Datasets

| File Name | Time Steps | Total Points | Spatial Dim |
|---|---|---|---|
| Simulation2D | 20 | 22,409 | 2D |
| Ambystoma 2D | 7 | 65,527 | 2D |
| DrosophilaEmbryo 3D | 3 | 94,805 | 3D |
| Human 3D | 100 | 500,000 | 3D |

### B.1. Simulation 2D Dataset

Simulation 2D Dataset is designed to simulate complex dynamics including morphological change, translation, population shifts, cell type specification, and spatio-temporal gene expression.

Let $t$ be the continuous time variable and $T = 10$ be the total number of integer time steps. The dynamics are defined as follows:

- **Geometry (Morphology):** The base shape is an equilateral triangle with a central hole. Both the triangle's scale $S(t)$ and the hole's radius $R_h(t)$ grow over time.

$$S(t) = 1.0 + 1.5 \sin\left(\frac{\pi(t-1)}{T-1}\right)$$

$$R_h(t) = 0.1 + 0.9\left(\frac{t-1}{T-1}\right)$$

- **Population (Cell Count):** The total cell count $N(t)$ follows a quadratic function, peaking at $t_{mid}$ with $N_{peak}$ cells.

$$N(t) = \max\left(0, \lfloor a(t - t_{mid})^2 + N_{peak}\rfloor\right)$$

- **Geometry (Translation):** The entire structure is translated along the X-axis by $X_{shift}(t)$, governed by a constant velocity $v_x = 2.0$.

- **Cell Types:** Three types (0, 1, 2) are defined by dynamic radii $R_0(t)$ and $R_1(t)$ expanding or contracting from the scaled base vertices $v_0(t)$ and $v_1(t)$.

- **Gene Expression:** Five genes ($G_0$ to $G_4$) are simulated based on coordinates and normalized time $t_{norm}$, derived from a Poisson distribution $\mathcal{P}(\lambda)$.

## B.2. Real-World Datasets

As described in Table 1 of the main paper:

- **Ambystoma 2D:** Ambystoma brain regeneration dataset (Wei et al., 2022).

- **Drosophila 3D:** Drosophila embryo development dataset (Wang et al., 2025). For the Drosophila 3D dataset, because only one slice remained in the final test set, we performed 5 runs experiments.

## B.3. Data-Specific Loss Adjustments

The final loss composition is adapted for specific datasets to reflect their distinct properties:

- **Drosophila 3D:** For this dataset, we disable the latent kinetic acceleration regularization term ($L_{kin}$), setting its weight $w_{kin} = 0$.

- **Human Motion 3D:** Since the Human Motion 3D (D-FAUST) dataset provides dense, paired geometric sequences without cell-specific attributes, we exclude the losses related to biological attributes and velocity alignment. Specifically, we disable the Trajectory Consistency Loss ($L_{traj}$, $w_{traj} = 0$) and all attribute-related losses ($L_{type}, L_{gene}, L_{exist}$), setting their corresponding weights to zero ($\gamma_{type} = 0, \gamma_{gene} = 0, \gamma_{exist} = 0$).

# C. Evaluation Metric Definitions

As promised in the main paper, we provide detailed definitions for the evaluation metrics.

## C.1. Geometric Fidelity (CD, OT)

To ensure the geometric metrics are robust to global translation, we first perform center alignment on both the predicted and ground-truth point clouds.

- **Chamfer Distance (CD):** A balanced version of the bidirectional Chamfer distance (Fan et al., 2017).

- **Sinkhorn Distance (OT):** Calculated using `geomloss.SamplesLoss("sinkhorn")` (Cuturi, 2013).

## C.2. Attribute Fidelity (CTA, SCC)

As our data is **unpaired**, we adopt an evaluation strategy based on **Nearest Neighbor (NN) Alignment**.

1. **Align:** We find the spatially closest neighbor $p_j \in S$ for each predicted point $\hat{p}_i \in \hat{S}$.

2. **Compute:**
    - **CTA:** Compares the predicted type of $\hat{p}_i$ with the true type of $p_j$.
    - **SCC:** Computes the Spearman's Correlation between the predicted gene vector $\hat{G}_i$ of $\hat{p}_i$ and the true gene vector $G_j$ of $p_j$.

## C.3. Population Fidelity (nP-MAE)

We introduce the normalized Population Mean Absolute Error (nP-MAE), which uses the mean relative error between the predicted count ($\hat{N}_t$) and the true count ($N_t$).

# D. Additional Qualitative

We provide comprehensive qualitative comparisons against state-of-the-art baseline methods across all datasets to visually demonstrate FlowCloud's superior capability in reconstructing complex spatiotemporal dynamics from sparse and unpaired data. These results validate the model's robustness and unified generative power. **Simulation 2D (Figure 13):** Comparison focuses on cell type reconstruction fidelity, complementing the gene expression results shown in the main paper. **Ambystoma**

**2D (Figures 14 and 15):** Detailed visualization of cell type distribution and specific gene expression profiles during brain regeneration, serving as comprehensive baseline comparisons for the biological case study. **Drosophila 3D (Figure 16):** Qualitative comparison of gene expression patterns across the 3D embryo, complementing the cell type results presented in the main paper. **Human Motion 3D (Figure 17):** Visual validation of geometric fidelity and smooth motion reconstruction over time, showcasing FlowCloud's extension capability beyond biological data.

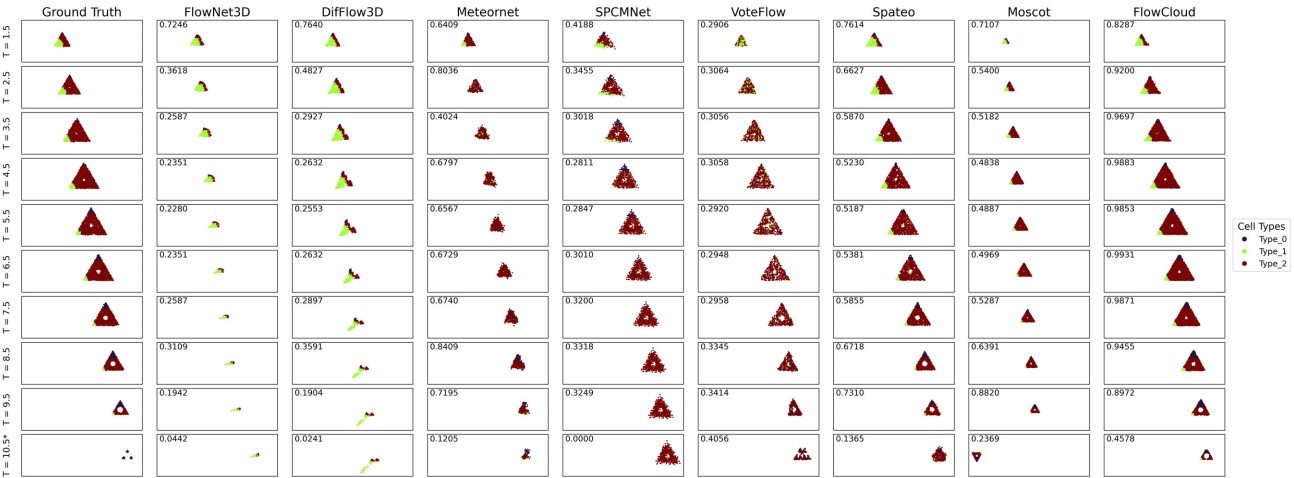

*Figure 13.* Qualitative Comparison of **Cell Type** Reconstruction on the **Simulation 2D** Dataset. This figure complements the main paper's Figure 5, which showed gene expression for this dataset. FlowCloud (rightmost column) demonstrates superior accuracy in capturing the correct spatial distribution of all cell types compared to baseline methods.

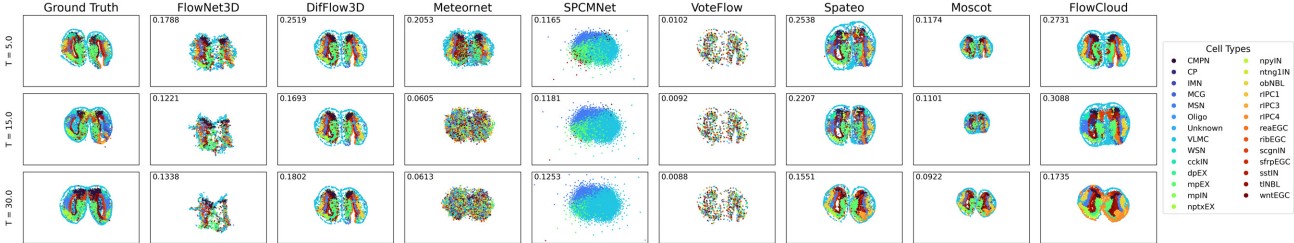

*Figure 14.* Qualitative Comparison of **Cell Type** Reconstruction on the **Ambystoma 2D** Dataset. This figure provides a detailed baseline comparison for the dataset featured in the case study (Figure 8 and 9). FlowCloud accurately reconstructs the complex spatial arrangement of cell types during regeneration.

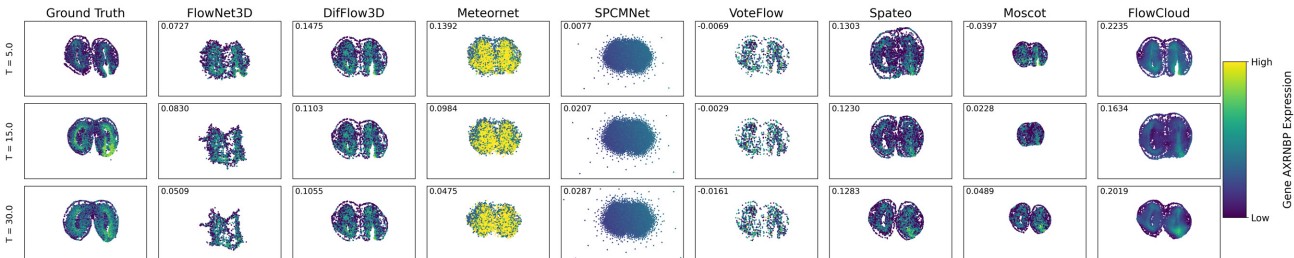

*Figure 15.* Qualitative Comparison of **Gene 0 Expression** on the **Ambystoma 2D** Dataset. This figure provides a detailed baseline comparison for a specific gene, complementing the holistic reconstruction shown in the main paper's Figure 7.

### D.1. Population Dynamics Analysis

The Figure 5 compares the predicted cell count trajectories (Total Count and individual Cell Types) from FlowCloud and the baseline Moscot (Klein et al., 2025) against Ground Truth over time. Figure 5 illustrates FlowCloud's ability to model complex **population dynamics** (e.g., cell birth/death/proliferation), which is essential in domains like developmental

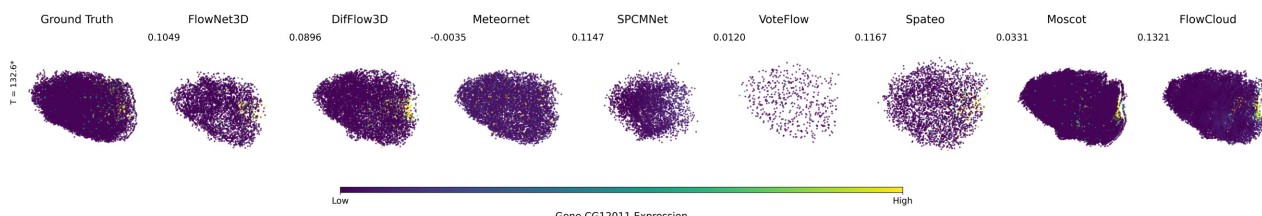

*Figure 16.* Qualitative Comparison of **Gene 0 Expression** on the **Drosophila 3D** Dataset. This figure complements the main paper's Figure 6, which showed cell types for this dataset. FlowCloud (rightmost column) successfully captures the spatiotemporal expression pattern of the selected gene.

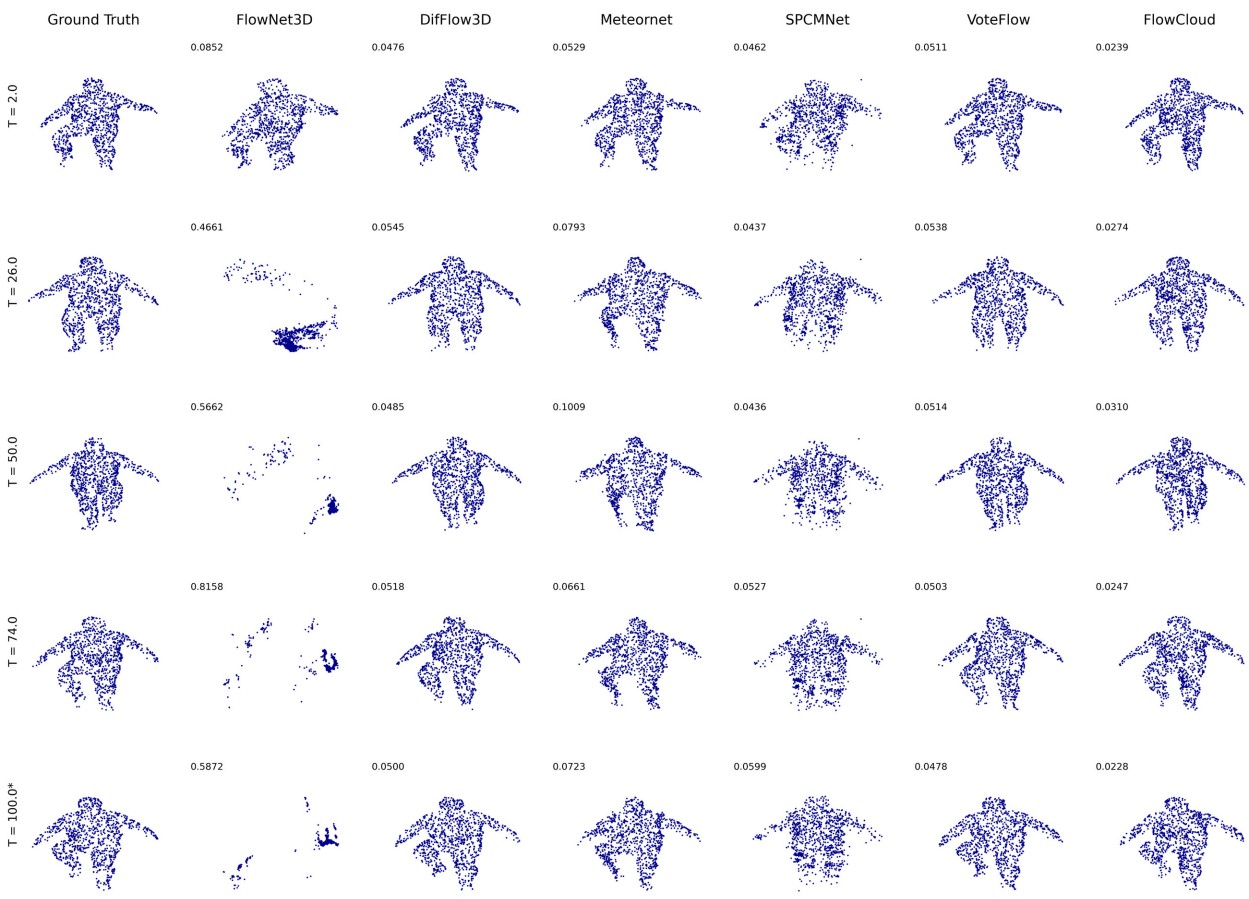

*Figure 17.* Qualitative Comparison of **Morphology** Reconstruction on the **Human 3D** (D-FAUST) Dataset ([Bogo et al.](), [2017](). FlowCloud (rightmost column) generates highly accurate and smooth geometric reconstructions of human motion compared to all baselines.

biology. The figure compares the ground truth cell count trajectories with predictions from FlowCloud and the discrete baseline Moscot ([Klein et al.](), [2025]()). The ground truth total cell count follows a complex, non-Markovian quadratic curve (peaking around time 5-6). FlowCloud accurately models this parabolic growth and subsequent decay (black dashed line). In contrast, the Optimal Transport-based method, Moscot (black dotted line), fails to capture the overall population change and maintains a largely flat total cell count. The model's superior performance is quantified by the Normalized Population Mean Absolute Error ($nP - MAE$), where FlowCloud achieves **0.2128**, significantly outperforming Moscot's **0.4931**. This verifies the efficacy of FlowCloud's integrated existence predictor and continuous dynamics kernel in handling non-rigid population shifts. FlowCloud successfully captures the continuous temporal shifts and magnitude changes for individual cell types (Type 0 and Type 2), further illustrating its grasp of the underlying generative dynamics.

# E. Statement on Experimental Fairness

To ensure a comprehensive and fair comparison, we adapted several baseline methods to address the specific challenges of our task (sparse, unpaired, multimodal data).

- **For Paired Point Cloud Methods (e.g., Scene Flow):** We used an autoregressive prediction strategy after simulating pairings via the Fused Gromov-Wasserstein (FGW) method (Vayer et al., 2020). As these methods lack native attribute prediction, we substituted the cell type and gene values from the **initial point** for all interpolated points.

- **For Methods with Continuous Process Recognition (e.g., SPCMNet):** We augmented their capabilities by adding corresponding prediction heads for cell types and genes and trained them using the **identical loss functions** as FlowCloud.

- **For Optimal Transport Methods (Spateo):** We only used it for **interpolation between two adjacent slices**, substituting attributes from the **source point cloud** (Qiu et al., 2024).

## E.1. Comparison with Continuous-Time Generative Baselines

## E.2. Robustness to Large-angle Rotations

To verify the robustness of FlowCloud against extreme spatial discrepancies, we introduced artificial global rotations $(30°, 45°, 60°)$ to the Ambystoma 2D dataset. Specifically, we manually rotated the unobserved training slices while keeping the test slices unchanged. We then applied PASTE (Zeira et al., 2022) as a standard preprocessing step for rigid alignment before feeding the data into FlowCloud.

As shown in Table 11, with PASTE preprocessing, FlowCloud maintains its baseline-level performance even under severe $60°$ rotations, effectively decoupling rigid spatial discrepancies from true non-linear developmental deformations.

*Table 11.* **Large-angle Rotation Robustness (Ambystoma 2D).**

| Rotation Angle | Pre-processing | CD ($\downarrow$) | SCC ($\uparrow$) |
|---|---|---|---|
| $0°$ (Original) | None | 0.2261 | 0.1960 |
| $30°$ | PASTE Alignment | 0.2264 | 0.1957 |
| $45°$ | PASTE Alignment | 0.2259 | 0.1962 |
| $60°$ | PASTE Alignment | 0.2265 | 0.1955 |

# F. Other Experiments and Discussions

## F.1. Trade-offs in Multimodal Integration

We evaluated the impact of the spatial distance weight ($w_{spatial}$) on the trajectory consistency loss. Setting $w_{spatial} = 0.9$ reflects our empirical hypothesis: macro-morphology acts as the anchor for short-term dynamics, but trajectories must still be guided by biological states.

Table 12 shows the results of an extreme setting ($w_{spatial} = 1.0$) where gene features are completely stripped from the trajectory consistency. Counterintuitively, focusing 100% on spatial distance degrades the geometric accuracy (CD increases from 0.137 to 0.212). In real development, structural deformations are driven by underlying gene expression changes. Ignoring gene features forces the model into naive spatial nearest-neighbor mappings, failing to capture true gene-driven cellular trajectories. This validates that multimodal integration is strictly necessary for reconstructing physical geometric evolution.

*Table 12.* **Ablation on Extreme $w_{spatial}$ (Sim 2D).**

| Metric | FlowCloud ($w_{spatial} = 0.9$) | FlowCloud ($w_{spatial} = 1.0$) |
|---|---|---|
| CD ($\downarrow$) | 0.1369 | 0.2123 |
| SCC ($\uparrow$) | 0.6273 | 0.5418 |

To provide a rigorous comparison with state-of-the-art continuous flow-matching models, we evaluated NicheFlow (Sakalyan et al., 2025). As shown in Table 13, NicheFlow's localized microenvironment approach struggles with macro-topological

deformations. FlowCloud's global context successfully synchronizes large-scale geometry and multimodal attributes, significantly outperforming NicheFlow in both spatial (CD) and transcriptomic (SCC) metrics.

*Table 13.* **Comparison with Latent ODE variants.**

| Dataset | Method | CD ($\downarrow$) | SCC ($\uparrow$) |
|---|---|---|---|
| Sim 2D | NicheFlow | 0.2352 | 0.4120 |
| | FlowCloud (Ours) | **0.1369** | **0.6273** |
| Amby 2D | NicheFlow | 0.3850 | 0.1240 |
| | FlowCloud (Ours) | **0.2261** | **0.1960** |

### F.2. Fine-Grained Ablation of the Composite Loss

To validate the design of our composite objective, we conducted a systematic ablation study on the Simulation 2D dataset by removing individual loss components.

As shown in Table 14, removing the global distribution alignment loss ($\mathcal{L}_{dist}$) causes a drastic increase in OT distance, confirming its critical role in aligning unpaired distributions. Without multimodal consistency ($\mathcal{L}_{multi}$), both SCC and CTA drop significantly. Finally, removing the trajectory smoothness regularization ($\mathcal{L}_{traj}$) destabilizes the dynamics, leading to less smooth trajectories and degraded attribute prediction.

*Table 14.* **Fine-Grained Ablation Study (Sim 2D).**

| Ablation Set | OT ($\downarrow$) | CD ($\downarrow$) | SCC ($\uparrow$) | CTA ($\uparrow$) |
|---|---|---|---|---|
| Full Loss (FlowCloud) | 0.015 | 0.137 | 0.627 | 0.897 |
| w/o $\mathcal{L}_{dist}$ | 0.082 | 0.485 | 0.584 | 0.812 |
| w/o $\mathcal{L}_{multi}$ | 0.098 | 0.518 | 0.475 | 0.542 |
| w/o $\mathcal{L}_{traj}$ | 0.026 | 0.228 | 0.512 | 0.695 |

## The Use of Large Language Models (LLMs)

During the preparation of this manuscript, we utilized Large Language Models (LLMs) as writing assistants. Specifically, we used Gemini Pro and DeepSeek to optimize the grammar, clarity, and readability of the text, and DeepSeek to assist with adjusting table formats. The role of the models was strictly limited to sentence rephrasing (to improve writing flow) and typographical error correction. All core scientific ideas, experimental design, analysis processes, and conclusions presented in this paper were independently conceived and developed by the human authors. We have carefully reviewed and edited all model-generated content, take full responsibility for the final manuscript, and ensure its scientific accuracy and originality.

