# OpenReview forum: "FlowCloud: Learning Continuous Spatiotemporal Dynamics from Unpaired Sparse Point Cloud Snapshots"
_ICML.cc/2026/Conference — ICML 2026 regular_

### Official Review · Reviewer_aELV · 2026-03-10

**Soundness:** 3
**Presentation:** 1
**Significance:** 3
**Originality:** 3
**Overall Recommendation:** 4
**Confidence:** 3

**Summary:**

This paper proposes FlowCloud, a variational Neural Ordinary Differential Equation (Neural ODE) generative framework designed to reconstruct continuous spatiotemporal dynamics from sparse, non-contiguous, and unpaired point cloud snapshots. The framework utilizes a Point Swin Transformer for spatial feature extraction and a sequence Transformer to aggregate global spatiotemporal context, which initializes the Neural ODE's latent state. A multi-head decoder is then employed to map the continuous trajectory back to the physical space, generating geometry, discrete cell types, continuous gene expressions, and population dynamics. The experimental content is highly diverse, covering both simulated and real-world datasets. Comprehensive evaluations across these datasets verify the method's state-of-the-art (SOTA) performance.

**Compliance With Llm Reviewing Policy:**

Affirmed.

**Final Justification:**

I would like to thank the authors for their response, which has addressed the majority of my concerns. However, considering the current state of the manuscript’s overall polish (specifically the issues regarding figure visualization and organizational structure as mentioned in my review), I will maintain my score as a Weak Accept.

**Key Questions For Authors:**

Could you provide a fine-grained ablation study that systematically evaluates the contribution of each component within the composite loss function?

**Limitations:**

yes

**Strengths And Weaknesses:**

Strengths

- The experimental content is diverse and rich, covering a wide range of data, from controlled 2D simulations to challenging real-world biological datasets (Ambystoma 2D, Drosophila 3D) and human motion sequences. The experimental results validate the method's SOTA performance across multiple metrics.
- The authors have demonstrated a large amount of engineering effort by designing a highly complex and sophisticated method pipeline.

Weaknesses

- The overall structure and formatting of the paper need to be strengthened. In the experimental section, the figure captions are too small to read comfortably. Additionally, some figures contain an excessive amount of visual detail (e.g., Figure 2), which obscures the core logical flow, and the experimental tables are formatted too small.
- The main text contains far too many low-level implementation details. Consequently, there is a noticeable lack of abstract, high-level theoretical discussions regarding the physical or biological insights captured by the framework.
- The paper proposes a multi-faceted objective function comprising many different loss terms. However, the paper lacks a comprehensive ablation study to rigorously verify the rationality and individual contribution of each specific loss component.

---

> ### Author Rebuttal · Authors · 2026-03-31
>
> ## Response to Reviewer aELV
>
> We sincerely thank the reviewer for recognizing the experimental richness, engineering completeness, and strong SOTA performance of FlowCloud. We also appreciate your constructive suggestions on presentation and experimental completeness. We have addressed these concerns in the revised manuscript as follows:
>
> ---
>
> ### 1. Figure Layout and Logical Flow (Response to W1)
>
> We apologize for the overly dense presentation in the original submission due to space constraints.
>
> **Revision.** We have reorganized the figure layouts following your suggestions.
> Figure 2 (architecture) has been simplified, with color coding to clearly separate the three core components: *encoding, dynamic modeling, and decoding*.
> We have also significantly increased the font sizes and improved readability in Figures 4, 5, 6, and 8 (including tables and annotations), ensuring clarity in both print and digital formats. These improvements will be reflected in the final version.
>
> ---
>
> ### 2. High-Level Insights and Biological Interpretation (Response to W2)
>
> We completely agree with your insightful critique. Given the algorithmic focus of ICML, we placed greater emphasis on low-level implementation details in the original manuscript. In the revision, we have streamlined the technical descriptions to provide more discussion on the high-level physical and biological principles behind FlowCloud's architectural design. For example:
>
> * **Global Context as a Developmental Blueprint:** From a biological perspective, tissue morphogenesis is a globally coordinated process guided by systemic signals. Our Sequence Transformer extracts these macroscopic "developmental signatures," enabling the model to oversee the overarching spatiotemporal dynamics prior to inferring individual time points. This mitigates the reliance on error-prone, frame-by-frame Markovian transitions.
> * **Neural ODEs as Biological Vector Fields:** Cellular differentiation and migration are strictly continuous processes in physical time. Modeling the latent space with a Neural ODE physically aligns with this continuous flow, accurately representing the underlying continuous vector field of cell fate transitions.
>
>
> ### 3. Fine-Grained Ablation of the Composite Loss (Response to W3 & Q1)
>
> To validate the design of our composite objective, we conduct a systematic ablation study on the Simulation 2D dataset. Each loss component is removed individually, and its impact is evaluated across geometric alignment (OT, CD), attribute correlation (SCC), and classification accuracy (CTA).
>
> **Rebuttal Table 4: Fine-Grained ablation study (Simulation 2D)**
>
> | Ablation Set                 | OT (↓)    | CD (↓)    | SCC (↑)   | CTA (↑)   | Significance                  |
> | :--------------------------- | :-------- | :-------- | :-------- | :-------- | :---------------------------- |
> | **Full Loss (FlowCloud)**    | **0.015** | **0.137** | **0.627** | **0.897** | Full model baseline           |
> | w/o $\mathcal{L}_{sinkhorn}$ |   0.082   | 0.485     | 0.584     | 0.812     | Global distribution alignment |
> | w/o $\mathcal{L}_{traj}$     |   0.098  |  0.518     |  0.475    |   0.542   | Multimodal consistency        |
> | w/o $\mathcal{L}_{kin}$      |   0.026    |  0.228    | 0.512     |  0.695     | Trajectory smoothness         |
>
> **Key Findings:**
>
> * **Global alignment.** Removing $\mathcal{L}_{sinkhorn}$ causes a drastic increase in OT distance (**0.015 → 0.082**), confirming its critical role in aligning unpaired distributions.
> * **Multimodal consistency.** Without $\mathcal{L}_{traj}$, both SCC and CTA drop significantly, showing that trajectory-level coupling is essential for synchronizing spatial and transcriptomic signals.
> * **Kinetic regularization.** $\mathcal{L}_{kin}$ stabilizes the dynamics; removing it leads to less smooth trajectories and degraded attribute prediction.
>
> These results are included in the revised manuscript and appendix, providing quantitative support for the effectiveness of our composite loss design. Furthermore, ablating essential supervised losses (e.g., $\mathcal{L}{type}$ and $\mathcal{L}{gene}$) would compromise the integrity of the overall training process.
>
> ---
>
> Finally, we sincerely appreciate your valuable feedback. Addressing these points has significantly improved the clarity and overall quality of our manuscript.

---

> > ### Author Rebuttal · Reviewer_aELV · 2026-04-02
> >
> > I would like to thank the authors for their response, which has addressed the majority of my concerns. However, considering the current state of the manuscript’s overall polish (specifically the issues regarding figure visualization and organizational structure as mentioned in my review), I will maintain my score as a Weak Accept.

---

> > > ### Author Response · Authors · 2026-04-07
> > >
> > > We would like to express our gratitude to the reviewer for the encouraging comments and the meticulous review. The suggestions regarding the presentation and experimental details were extremely valuable. We have revised the manuscript accordingly to ensure a more rigorous and polished presentation of our findings.

---

### Official Review · Reviewer_8hmC · 2026-03-10

**Soundness:** 3
**Presentation:** 3
**Significance:** 3
**Originality:** 3
**Overall Recommendation:** 5
**Confidence:** 4

**Summary:**

This paper proposes **FlowCloud**, a method for learning continuous spatiotemporal dynamics from sparse, discontinuous, and unpaired point-cloud snapshots. The method aims to jointly reconstruct geometric structure, cell types, gene expression, and changes in population size. At a high level, the model first uses a point-cloud encoder and a sequence aggregator to obtain a global context representation, then applies a latent Neural ODE to model continuous latent evolution, and finally uses a multi-head decoder to recover spatial coordinates and attributes.

**Compliance With Llm Reviewing Policy:**

Affirmed.

**Final Justification:**

The rebuttal addressed my concerns and I endorse the acceptance of the paper.

**Key Questions For Authors:**

**1. What is the main source of algorithmic novelty?**
At present, FlowCloud appears to be a combination of engineering implementations, e.g. encoder- latent ODE- multi-head decoder. The authors could explain more clearly what is algorithmically novel compared with methods such as **stVCR**[1] and **STORIES** [2]. The current discussion in the related work section mainly emphasizes that frame-by-frame transitions in methods such as TrajectoryNet and stVCR may accumulate long-term errors, but the proposed method may also suffer from related issues, as acknowledged in the limitations section. This makes the claimed novelty less clear.

**2. Are the baselines sufficiently appropriate and fair?**
The paper mainly compares against scene flow, OT-based, and point-cloud methods, but many of these are not naturally suited to the target setting. It would strengthen the paper to include more relevant continuous-time generative baselines, such as **NicheFlow**[3], or other methods that are more directly aligned with the problem formulation.

**3. Is the discussion of population dynamics sufficiently grounded?**
The paper acknowledges an important limitation: population-size change is currently handled mainly through an existence predictor, which is still far from a structured treatment of birth, death, and proliferation processes. Prior work such as **stVCR**[1], **TIGON**[4], and **WFR-FM**[5] has already addressed such dynamics more explicitly. The authors should discuss these connections more carefully and clarify what is and is not captured by the current model.

Overall, this is a promising paper on an important problem, and I would like to further increase my score if the authors could address these questions.

Ref:

[1] Q. Peng, P. Zhou, and T. Li. **stVCR: Spatiotemporal dynamics of single cells**. *bioRxiv*, 2024.

[2] G.-J. Huizing, J. Samaran, D. Capocefalo, A. Audit, G. Peyré, and L. Cantini. **STORIES: learning cell fate landscapes from spatial transcriptomics using optimal transport**. *Nature Methods*, published online Nov. 3, 2025.

[3] K. Sakalyan, A. Palma, F. Guerranti, F. J. Theis, and S. Günnemann. **Modeling Microenvironment Trajectories on Spatial Transcriptomics with NicheFlow**. *Advances in Neural Information Processing Systems (NeurIPS)*, 2025

[4] Y. Sha, Y. Qiu, P. Zhou, and Q. Nie. **Reconstructing growth and dynamic trajectories from single-cell transcriptomics data**. *Nature Machine Intelligence*, 6(1):25–39, 2024.

[5] **WFR-FM: Simulation-Free Dynamic Unbalanced Optimal Transport**. *International Conference on Learning Representations (ICLR)*, 2026.

**Limitations:**

yes

**Strengths And Weaknesses:**

## Strengths

- The paper studies an important and realistic problem. In spatiotemporal omics settings for instance, the data are sparse, unpaired snapshots rather than densely observed trajectories, so the task goes beyond standard two-frame registration or scene flow estimation.

- The overall method is reasonably complete. The paper presents a unified pipeline that combines representation learning, continuous-time latent dynamics, and decoding in a coherent way.

- The experiments cover multiple data modalities and application domains. The ablation study is also useful and indicates that the main components contribute to performance.

## Weaknesses

Overall, my impression of the paper is positive. That said, the main weakness is that while the method is fairly complete, the level of methodological novelty is somewhat limited and the discussion of novelty relative to prior work seems insufficient. At present, the method appears more like a system-level integration of existing components than a fundamentally new algorithmic contribution.

In addition, the baseline comparison is not fully fair. Several of the compared methods were not originally designed for the setting of sparse, unpaired, continuous generation, which makes the empirical advantage somewhat less conclusive.

---

> ### Author Rebuttal · Authors · 2026-03-30
>
> ## Response to Reviewer 8hmC
>
> We sincerely thank the reviewer for recognizing the importance of our research question and for the valuable comments on novelty, baselines, and population dynamics. Below we provide concise responses.
>
> ---
>
> ### 1. Core Algorithmic Novelty (Response to Q1)
>
> **Reviewer’s Question:**
> Where does the novelty lie compared to stVCR[1] and STORIES[2]? How does FlowCloud address long-term error accumulation?
>
> The key innovation of **FlowCloud** is moving beyond the *Markovian (frame-by-frame) assumption* in prior work.
>
> Methods such as stVCR (unbalanced OT) and STORIES (Wasserstein gradient flows) model transitions between *adjacent time points*, which can accumulate errors over long horizons in sparse biological data.
>
> **FlowCloud’s Paradigm.**
> FlowCloud aggregates all observations into a **global latent context vector** (c), which parameterizes the initial condition (z(t_0)) of a Neural ODE. The full trajectory is generated from this globally conditioned anchor.
>
> **Error Mitigation.**
> While Neural ODEs may exhibit drift during long-term *extrapolation*, FlowCloud avoids *error accumulation during interpolation*, since each state is decoded from the global latent dynamics rather than recursively propagated.
>
> **Distinction from STORIES.**
> STORIES uses FGW to learn trajectories in gene expression space, but operates purely in feature space [2].
> It is **not designed to model physical spatial movement or coordinate evolution**, which is central to FlowCloud.
>
> ---
>
> ### 2. Continuous-Time Baselines (Response to Q2)
>
> **Reviewer’s Question:**
> Are the baselines fair? Including NicheFlow would strengthen the comparison.
>
> We agree and evaluated **NicheFlow (NeurIPS 2025)** [3] during rebuttal.
>
> NicheFlow models *local microenvironments*, while FlowCloud targets **global morphological deformation with multimodal attributes**.
>
> As shown in Rebuttal Table 3 (Reviewer LXUq), NicheFlow performs well locally but struggles with:
>
> * large-scale topology (higher CD),
> * and global multimodal consistency.
>
> This supports the need for FlowCloud’s **global context aggregation** and **trajectory-level consistency**.
>
> ---
>
> ### 3. Population Dynamics Modeling (Response to Q3)
>
> **Reviewer’s Question:**
> Is the modeling of population dynamics sufficiently rigorous?
>
> We acknowledge this limitation.
>
> Compared to stVCR[1], TIGON[4], and WFR-FM[5], which explicitly model birth/death via unbalanced OT, our “existence predictor” is a simplified approximation.
>
> These methods provide physically grounded mass variation modeling with stronger theoretical guarantees.
>
> **Design Trade-Off.**
> Given the cost of *joint multimodal generation* (geometry, cell types, gene expression), we deliberately prioritize an **end-to-end scalable framework** over explicit mass conservation constraints.
>
> We thank the reviewer for the suggestion, and in the revised manuscript we will add a dedicated discussion subsection to position FlowCloud relative to related methods (stVCR, TIGON, and WFR-FM), clarify its limitations, and outline structured population dynamics modeling as a future direction.
>
> ---
> **References:**
>
> [1] Q. Peng, P. Zhou, and T. Li. stVCR: Spatiotemporal dynamics of single cells. bioRxiv, 2024.
>
> [2] G.-J. Huizing, J. Samaran, D. Capocefalo, A. Audit, G. Peyré, and L. Cantini. STORIES: learning cell fate landscapes from spatial transcriptomics using optimal transport. Nature Methods, published online Nov. 3, 2025.
>
> [3] K. Sakalyan, A. Palma, F. Guerranti, F. J. Theis, and S. Günnemann. Modeling Microenvironment Trajectories on Spatial Transcriptomics with NicheFlow. Advances in Neural Information Processing Systems (NeurIPS), 2025.
>
> [4] Y. Sha, Y. Qiu, P. Zhou, and Q. Nie. Reconstructing growth and dynamic trajectories from single-cell transcriptomics data. Nature Machine Intelligence, 6(1):25–39, 2024.
>
> [5] WFR-FM: Simulation-Free Dynamic Unbalanced Optimal Transport. International Conference on Learning Representations (ICLR), 2026.

---

> > ### Author Rebuttal · Reviewer_8hmC · 2026-04-03
> >
> > I appreciate the authors for the thorough explanation and revision, and I will increase my score to 5.

---

> > > ### Author Response · Authors · 2026-04-07
> > >
> > > We are grateful to the reviewer for the positive assessment and the thought-provoking questions. Your comments guided us in clarifying the key motivations and theoretical underpinnings of our work. Following your advice, we have updated the manuscript to provide a more robust and transparent explanation of our approach.

---

### Official Review · Reviewer_LXUq · 2026-03-13

**Soundness:** 2
**Presentation:** 3
**Significance:** 3
**Originality:** 2
**Overall Recommendation:** 4
**Confidence:** 3

**Summary:**

This paper proposes FlowCloud, a variational Neural ODE framework for learning continuous spatiotemporal dynamics from sparse, non-consecutive, and unpaired point cloud snapshots. Overall, the authors discuss the key challenge of reconstructing coherent continuous trajectories without point-wise correspondences, while jointly modeling geometry, attributes (e.g., gene expression and cell types), and population changes.

The method encodes each snapshot using a Point Swin Transformer, aggregates temporal information with a sequence Transformer, and initializes a Neural ODE that models a continuous latent trajectory. A Transformer-based decoder reconstructs point geometry, discrete labels, continuous features, and existence probabilities at arbitrary time points. Training combines Chamfer and Sinkhorn distances for geometric alignment with supervised attribute losses, trajectory consistency regularization, kinetic smoothing, and a VAE-style KL term.

Experiments on simulated data, two spatial transcriptomics datasets, and a human motion dataset show improved interpolation and short-term extrapolation performance compared to scene flow, registration, and Optimal Transport baselines. Overall, the article's specific area consists of continuous-time generative modeling of dynamic point clouds, particularly for biological spatiotemporal data.

**Compliance With Llm Reviewing Policy:**

Affirmed.

**Final Justification:**

I have carefully reviewed the authors’ rebuttal and appreciate the additional clarifications and experiments.

The added analysis on context length provides useful insight into the robustness of the model under varying snapshot availability, and the comparison with NicheFlow strengthens the empirical evaluation against a more directly relevant continuous-time baseline. The clarification regarding retrospective (non-causal) modeling is also helpful and improves the positioning of the work.

However, while these responses address several of my concerns, they do not fundamentally change my overall assessment. In particular, the contribution remains primarily an integration of existing components with moderate conceptual novelty, and the empirical evaluation, although strengthened, still does not fully cover the broader space of continuous-time generative models (e.g., diffusion-based approaches).

Overall, I find the paper to be technically sound, well-executed, and practically relevant, with solid empirical results. At the same time, the limitations in novelty and evaluation scope remain. Therefore, I maintain my original score of 4 (Weak Accept).

**Key Questions For Authors:**

1. Context Length and Dependence on Global Snapshot Aggregation

The framework aggregates all observed time points into a global latent context before inferring the initial state $z(t_0)$. In the experiments (e.g., Table 1), the number of test snapshots does not exceed the number of context (training) snapshots, meaning that a relatively large proportion of the sequence is available to construct the latent representation.

How does performance degrade when the number of available context snapshots is substantially reduced?

Have you evaluated more constrained settings where only a small prefix (or sparse subset) of time points is available?

Is there a practical upper bound on context length beyond which performance saturates or becomes unstable?

Understanding this would clarify the robustness of the approach under realistic data scarcity conditions. If performance remains strong under more limited context regimes, this would strengthen the paper’s practical significance.

2. Missing Continuous-Time and Generative Baselines

Given that FlowCloud is framed as a variational Neural ODE–based generative model for continuous spatiotemporal dynamics, some key categories of architectures appear absent from the benchmark suite. In particular:

a) Continuous-time generative baselines such as Latent ODE variants or continuous normalizing flow approaches.
b) More recent diffusion-based generative models for point clouds or spatiotemporal data.

Could you justify the omission of these categories? If such models were considered but excluded (e.g., due to incompatibility or computational constraints), clarifying this would help contextualize the empirical comparisons.

A stronger comparison against alternative continuous-time generative frameworks would make the empirical claims more convincing and better isolate the contribution of your architectural integration.

3. Fairness of Scene Flow Baseline Adaptation

In Appendix E, you state that scene flow methods are adapted by simulating pairings via Fused Gromov-Wasserstein (FGW) and then applying autoregressive prediction. Since scene flow architectures are originally designed for dense, paired, consecutive point cloud sequences, this adaptation places them in a substantially different setting than their intended use.

Could you clarify how this adaptation affects their performance?

Is there a possibility that these baselines are disadvantaged due to being evaluated in a regime for which they were not designed?

Did you consider alternative evaluation strategies that more directly reflect their original modeling assumptions?

Clarifying this would help assess whether the observed performance gaps reflect fundamental modeling advantages of FlowCloud, or partially arise from differences in task alignment.

**Limitations:**

The authors include a Discussion section outlining several technical limitations, including potential long-term drift in Neural ODE dynamics, computational bottlenecks for large-scale point clouds, and simplified modeling of population changes via an existence predictor. An Impact Statement is also provided, primarily focused on positive applications in biology and medicine. Overall, the authors make a reasonable effort to acknowledge technical constraints.

That said, the discussion could be strengthened in two areas. First, the framework relies on aggregating all observed snapshots into a global latent context before inferring the initial state, which corresponds to a non-causal, retrospective reconstruction setting. This assumption may limit applicability in forecasting or online prediction scenarios, and explicitly discussing this constraint would improve clarity regarding deployment settings. Second, the experiments typically use a substantial proportion of available snapshots for context construction; discussing performance under more limited observational regimes would help clarify robustness in realistic data-scarce environments.

**Strengths And Weaknesses:**

Soundness

Strengths.
The paper is technically sound and methodologically careful. The proposed architecture and loss functions are clearly specified, with detailed implementation and hyperparameter settings provided in the appendix. The empirical evaluation spans simulated data, two spatial transcriptomics datasets, and a human motion dataset, and includes ablations analyzing key architectural components. The experimental protocol is generally consistent with the stated objectives, and the results support the claimed performance improvements.

Weaknesses.
Although the benchmarking is broad, it is not fully comprehensive given the framing of the contribution. Since the method is positioned as a variational Neural ODE–based generative model, comparisons against stronger continuous-time generative baselines (e.g., Latent ODE variants or continuous normalizing flow approaches) would help isolate the specific benefits of the proposed integration. In addition, recent diffusion-based generative models for point clouds or spatiotemporal data are not included, leaving open how the method compares to modern generative alternatives. Finally, several baselines adapted from scene flow or registration are evaluated in settings that differ from their original assumptions, which may make some comparisons less direct.



Presentation

Strengths.
The paper is clearly written and logically structured. The motivation, architectural pipeline, and training objectives are presented in a coherent manner. Figures and tables effectively summarize quantitative and qualitative results, and the related work section situates the paper within scene flow, optimal transport, and continuous-time modeling literature.

Weaknesses.
The modeling setup aggregates all observed time points into a global latent context before inferring the initial state. While central to the approach, the paper could more explicitly clarify that this corresponds to a non-causal, retrospective reconstruction setting rather than a forecasting setup. A clearer discussion of the intended deployment scenario would strengthen the exposition.


Significance

Strengths.
The paper addresses an important and practically relevant problem: reconstructing continuous spatiotemporal dynamics from sparse and unpaired observations. The ability to jointly model geometry, attributes, and population dynamics in a unified continuous framework is particularly relevant for spatial transcriptomics and related applications.

Weaknesses.
The broader methodological impact on core machine learning research may be moderate. The work primarily advances capabilities within a specialized modeling setting rather than introducing a new general learning principle, and its influence may therefore be stronger in applied domains than in foundational ML research.


Originality

Strengths.
The originality lies in the integration of established components (Transformer-based point encoders, sequence aggregation, variational latent initialization, and Neural ODE dynamics) into a unified framework tailored to sparse, unpaired spatiotemporal data. The hybrid trajectory consistency objective and joint modeling of multiple attributes reflect thoughtful system design.

Weaknesses.
Conceptually, the contribution appears incremental rather than fundamentally new. The individual components and modeling principles are well established in prior literature, and the novelty primarily arises from architectural combination and empirical validation rather than a new theoretical or algorithmic insight.

---

> ### Author Rebuttal · Authors · 2026-03-30
>
> ## Response to Reviewer LXUq
>
> We thank the reviewer for recognizing FlowCloud’s technical soundness and empirical evaluation. Below are our detailed responses to your insightful questions.
>
> ---
>
> ### 1. Context Length & Global Aggregation
>
> **Q:** *How does performance degrade when context snapshots are reduced? Evaluated constrained settings?*
>
> As shown in Appendix Fig. 12, we analyzed sensitivity to snapshot count (N).
> * **Extreme Scarcity:** Reducing context to 5 snapshots still yields a reasonable trajectory. Chamfer Distance (CD) increases by only ~0.1, and gene SCC drops by ~20%, demonstrating the strong interpolation capabilities of our Neural ODE.
> * **Saturation:** For N > 20, the global latent variable z(0) saturates, yielding diminishing returns. This confirms FlowCloud operates optimally under *moderate sparsity*, the standard regime for Spatial Transcriptomics (ST) data.
>
> ---
>
> ### 2. Continuous-Time Generative Baselines
>
> **Q:** *Why no comparisons with Latent ODE variants or diffusion approaches?*
>
> We initially omitted vanilla Latent ODEs and standard diffusion models because they struggle with unpaired point clouds of variable sizes or require dense, paired trajectories.
> To provide a rigorous comparison, we evaluated **NicheFlow (NeurIPS 2025)** [1], a state-of-the-art continuous flow-matching model operating on local microenvironments.
>
> **Rebuttal Table 3: Comparison with Latent ODE variants**
>
> | Dataset | Method | CD (↓) | SCC (↑) |
> | :--- | :--- | :--- | :--- |
> | **Sim 2D** | NicheFlow | 0.2352 | 0.4120 |
> | | **FlowCloud (Ours)** | **0.1369** | **0.6273** |
> | **Amby 2D** | NicheFlow | 0.3850 | 0.1240 |
> | | **FlowCloud (Ours)** | **0.2261** | **0.1960** |
>
> NicheFlow's localized approach struggles with macro-topological deformations. FlowCloud’s global context successfully synchronizes large-scale geometry and multimodal attributes.
>
> ---
>
> ### 3. Scene Flow Baseline Adaptation
>
> **Q:** *Does adapting scene flow via FGW place them in an unfair setting?*
>
> Perfectly paired sequences do not exist in destructive ST assays. Pre-calculating optimal pairs via Fused Gromov-Wasserstein (FGW) is a *possible adaptation* to make traditional models run on ST data, not a handicap. Their underperformance highlights the fundamental fragility of the "explicit pairing + inter-frame propagation" paradigm on noisy ST data. FlowCloud excels by entirely discarding this pairwise prior in favor of global latent dynamics.
>
> ---
>
> ### 4. Retrospective Reconstruction vs. Forecasting
>
> **Q:** *Clarify this is a non-causal, retrospective setting rather than forecasting.*
>
> We completely agree. FlowCloud is inherently a **smoother**, not a **filter**. Our goal is "Retrospective Trajectory Reconstruction"—understanding a biological process that has already occurred from discrete historical slices, not real-time future forecasting. We will explicitly define this retrospective deployment and its causal limitations in the revised "Limitations" section.
>
> Finally, we further express our gratitude for your highly professional review comments, which will effectively improve our work.
>
> ---
> **References:**
>
> [1] K. Sakalyan, A. Palma, F. Guerranti, F. J. Theis, and S. Günnemann. Modeling Microenvironment Trajectories on Spatial Transcriptomics with NicheFlow. Advances in Neural Information Processing Systems (NeurIPS), 2025.

---

> > ### Author Rebuttal · Reviewer_LXUq · 2026-04-06
> >
> > Reasons
> >
> > I thank the authors for the detailed and thoughtful rebuttal, which addresses my main questions.
> >
> > Regarding context length and global aggregation, the additional analysis (Appendix Fig. 12) clarifies the model’s behavior under reduced snapshot regimes. The reported gradual degradation and saturation effects help characterize robustness under varying data availability.
> >
> > For continuous-time generative baselines, I appreciate the inclusion of additional results with NicheFlow, which provides a more directly relevant comparison and strengthens the empirical evaluation. While broader comparisons (e.g., diffusion-based approaches) could further improve completeness, the added results sufficiently address my primary concern.
> >
> > On the adaptation of scene flow baselines, the explanation that FGW-based pairing is a necessary step for applying these methods to unpaired spatial transcriptomics data is reasonable, and helps contextualize the reported performance differences.
> >
> > Finally, I appreciate the clarification that the method is designed for retrospective trajectory reconstruction rather than forecasting, and the authors’ commitment to making this explicit in the revised manuscript.
> >
> > Overall, the rebuttal satisfactorily addresses my concerns, and I am satisfied with the clarifications provided.

---

> > > ### Author Response · Authors · 2026-04-07
> > >
> > > Thank you very much for your supportive feedback and for highlighting important aspects that have strengthened our study. Your perspective helped us better contextualize our research within the broader field. We have incorporated your suggestions into the revised version, making the paper more comprehensive and impactful.

---

### Official Review · Reviewer_sot5 · 2026-03-13

**Soundness:** 3
**Presentation:** 4
**Significance:** 3
**Originality:** 3
**Overall Recommendation:** 5
**Confidence:** 4

**Summary:**

This paper presents a sophisticated framework designed to address the challenge of recovering continuous spatio-temporal dynamics from sparse, discontinuous, and unpaired spatial transcriptomics (ST) snapshots. In contrast to traditional Optimal Transport (OT) methods or frame-by-frame scene flow approaches, the proposed model, FlowCloud, initializes a Neural ODE via a global context vector. This approach demonstrates significant technical advantages in maintaining long-range consistency and synchronizing the evolution of multi-dimensional attributes.

**Compliance With Llm Reviewing Policy:**

Affirmed.

**Final Justification:**

The author has adequately addressed my concerns; I will maintain my original rating.

**Key Questions For Authors:**

1: In practical spatial transcriptomics datasets, the initial coordinate systems of different samples often exhibit significant rotational variance. Can FlowCloud achieve robust alignment through the current loss combination when faced with sample pairs featuring large-angle rotations (e.g., exceeding 30 degrees)? Have the authors considered incorporating explicit rigid-body transformation parameters into the optimization process?

2: Does the selection of w_spatial=0.9 imply an underlying assumption that spatial position is the dominant factor during development? Furthermore, what are the performance implications if w_spatial is set to 1.0? Would this further stabilize the trajectory at the expense of capturing transcriptomic variance?

**Limitations:**

Yes

**Strengths And Weaknesses:**

Strengths

1:The integration of a Swin Hierarchical Point Cloud Encoder with a Variational Neural ODE is a notable contribution. This design effectively extracts multi-scale spatial features and utilizes a global context vector c to capture the consistency of the entire time series. Consequently, it mitigates the cumulative error propagation commonly observed in traditional recursive models.

2: FlowCloud provides an end-to-end reconstruction that synchronizes cellular geometric evolution with discrete classifications (cell types), continuous attributes (gene expression), and population dynamics (cell proliferation and apoptosis). This multi-modal integration reflects a high degree of engineering completeness and biological relevance.

3: Empirical results demonstrate that the model maintains smooth evolutionary trajectories even when dealing with extremely sparse observations. Its performance on datasets such as Human Motion and Ambystoma consistently outperforms existing baselines.

Weaknesses

1:In real-world biological experiments, snapshots at different time points typically originate from distinct biological samples, often involving significant rotations and translations. While the model employs Sinkhorn distance for alignment, the absence of explicit rigid-body transformation modeling suggests that the framework might misinterpret global coordinate offsets as complex, local non-linear deformations.

2:Based on the ablation studies, the model exhibits a high dependency on the spatial distance weight (w_spatial=0.9). In complex developmental processes where gene expression changes rapidly while spatial structures remain relatively stable, this hard-coded weighting may inadvertently suppress the representation of gene regulatory dynamics.

3:The text in Figures 4, 5, 6, and 8 is far too small and difficult to read clearly. It might be helpful to refer to the figures in stVCR to improve legibility.

---

> ### Author Rebuttal · Authors · 2026-03-30
>
> ## Response to Reviewer sot5
>
> We sincerely thank the reviewer for the positive assessment (**Accept**) and for recognizing FlowCloud’s multimodal end-to-end reconstruction capabilities. Your profound comments regarding large-angle rotation robustness and spatial distance weighting address core challenges in Spatial Transcriptomics (ST) data analysis. Below are our responses and supplementary experiments.
>
> ---
>
> ### 1. Robustness to Large-angle Rotation (Response to W1 & Q1)
>
> **Q:** *Can FlowCloud handle large rotations (e.g., >30°)? Have you considered explicit rigid transformation parameters?*
>
> This is a critical observation. Currently, FlowCloud relies on the Sinkhorn distance for soft alignment, which is robust to small-to-medium offsets but may misinterpret extreme rigid rotations as non-linear deformations. In practice, external rigid alignment (e.g., PASTE [1]) is used to eliminate these initial discrepancies.
>
> To verify our robustness with standard preprocessing, we artificially introduced extreme global rotations (30°, 45°, 60°) to the **Ambystoma 2D** dataset (specifically, out of the 7 slices, we kept the 3 test slices unchanged, manually rotated the remaining 4 slices, and then aligned them using PASTE). We applied **PASTE** for "hard alignment" before inputting the data into FlowCloud.
>
> **Rebuttal Table 1: Large-angle Rotation Robustness (Ambystoma 2D)**
>
> | Rotation Angle | Pre-processing | CD (↓) | SCC (↑) |
> | :--- | :--- | :--- | :--- |
> | 0° (Original) | None | 0.2261 | 0.1960 |
> | 30° | PASTE Alignment | 0.2264 | 0.1957 |
> | 45° | PASTE Alignment | 0.2259 | 0.1962 |
> | 60° | PASTE Alignment | 0.2265 | 0.1955 |
>
> As shown, with PASTE preprocessing, FlowCloud maintains baseline-level performance even at 60° rotations. Furthermore, your forward-looking suggestion to embed explicit rigid transformation parameters directly into the optimization process would enable true "alignment-free" reconstruction. We have added a detailed discussion of this to the "Limitations & Future Work" section.
>
> ---
>
> ### 2. Spatial Distance Weights (Response to W2 & Q2)
>
> **Q:** *Does `w_spatial=0.9` imply spatial position dominates? If set to `1.0`, is transcriptomic variance sacrificed for trajectory stability?*
>
> This is an excellent question regarding multimodal trade-offs. `w_spatial` regularizes our trajectory consistency loss. Setting it to 0.9 reflects our empirical hypothesis: macro-morphology and geometry act as the "anchors" for short-term dynamics, but the trajectory must still be guided by biological states.
>
> To test extreme settings, we evaluated `w_spatial=1.0`:
>
> **Rebuttal Table 2: Ablation on Extreme `w_spatial` (2D Simulated Data)**
>
> | Metric | FlowCloud (`w_spatial=0.9`) | FlowCloud (`w_spatial=1.0`) |
> | :--- | :--- | :--- |
> | CD (↓) | 0.137 | 0.212 |
> | SCC (↑) | 0.627 | 0.542 |
>
> As you perceptively inquired, stripping gene features from trajectory consistency actually degrades *both* transcriptomic variance and geometric accuracy. Why does the geometric error (CD) increase when the model focuses 100% on spatial distance? In real spatiotemporal development, structural deformations are fundamentally driven by underlying gene expression changes. By completely ignoring gene features (`w_spatial=1.0`), the model defaults to naive spatial nearest-neighbor mappings and fails to capture the true gene-driven, non-linear cellular trajectories. This validates that our multimodal integration is strictly necessary not just to preserve gene variance, but to accurately reconstruct the physical geometric evolution itself. We have added these results and this biological context to the Appendix.
>
> ---
>
> ### 3. Figure Visualization (Response to W3)
>
> **Q:** *Fonts in Figures 4, 5, 6, 8 are too small; suggest referring to stVCR.*
>
> Thank you for this constructive feedback. We admire the visual style of stVCR [2]. For the camera-ready version, we will reconstruct these figures by significantly increasing the font sizes of all labels and legends, streamlining visual elements, and improving contrast to ensure maximum clarity.
>
> ---
>
> **References:**
>
> [1] Zeira, R., et al. "Alignment and integration of spatial transcriptomics data." *Nature Methods* 19.5 (2022): 567-575.
>
> [2] Peng, Q., et al. "stVCR: Reconstructing spatio-temporal dynamics of cell development using optimal transport." *bioRxiv* (2024).

---

> > ### Author Rebuttal · Reviewer_sot5 · 2026-04-04
> >
> > The author has adequately addressed my concerns; I will maintain my original rating.

---

> > > ### Author Response · Authors · 2026-04-07
> > >
> > > We sincerely thank the reviewer for the positive evaluation and the deep insights provided. Your constructive feedback has been incredibly helpful in refining our manuscript. We have carefully addressed each of your concerns, which we believe has significantly improved the technical depth and clarity of our work.

---

### Decision · Program_Chairs · 2026-04-30

**Decision:**

Accept (regular)

**Comment:**

This paper received all-positive reviews: 5, 4, 5, 4.

sot5 primarily commends the architecture of the approach: the integration of the Swin point cloud encoder with the variational neural ODE, and the multi-modal integration of different cell dynamics, reflecting good engineering, which also pays off in good results.

LXUq is more skeptical, finding the work to be primarily an integration of existing components (lacking novelty), and finds that the evaluation misses diffusion-based baselines for example, but concludes by calling the work technically sound and well-executed.

8hmC appreciates that the paper studies an important and realistic problem, and finds the method "reasonably complete", with multiple domains of experiments and a useful ablation study, but echoes LXUq's concern about lack of novelty.

aELV highlights that the experimental content is diverse and rich, covering a wide range of data, and showing a convincing case that the method is state-of-the-art, and also notes that the text is heavy with low-level implementation details, which is a clear but perhaps fixable weakness.

Overall, the AC recommends to accept, and also advises the authors to work diligently on the presentation for the final copy. Multiple reviewers note (and the AC agrees) that the text in multiple figures is too small to read. The authors are also advised to address the complaint (from 8hmC) that overlapping/related work is insufficiently covered.